# OKBench: Democratizing LLM Evaluation with Fully Automated, On-Demand Open Knowledge Benchmarking

## Abstract

Knowledge-intensive question answering is central to large language models (LLMs) and is typically assessed using static benchmarks derived from sources like Wikipedia and textbooks. However, these benchmarks fail to capture evolving knowledge in a dynamic world, and centralized curation struggles to keep pace with rapid LLM advancements. To address these drawbacks, we propose OpenKnowledgeBench (**OKBench**), a fully automated framework for generating high-quality, dynamic knowledge benchmarks on demand. Focusing on the news domain where knowledge updates daily, **OKBench** is an agentic framework that automates the sourcing, creation, validation, and distribution of benchmarks. Our approach democratizes benchmark creation and facilitates thorough evaluation of retrieval-augmented methods by reducing overlap with pretraining data. We evaluate our framework on multiple open-source and proprietary LLMs of various sizes and configurations, both with and without retrieval over freshly generated knowledge. Our results reveal distinct model behaviors when confronted with new information and highlight how retrieval narrows the performance gap between small and large models. These findings underscore the importance of evaluating LLMs on evolving knowledge benchmarks.[1]

## 1 Introduction

One of the most common uses of large language models (LLMs) is for answering knowledge-intensive questions. However, this task is challenging as factual knowledge in the real world evolves rapidly. Well-trained models can quickly become outdated (Li et al., 2024), raising the need for continual model updates (Liska et al., 2022) or improved retrieval-augmented generation (RAG) techniques (Lewis et al., 2020). At the same time, the lack of transparency in training data makes it difficult to assess how recent a model's knowledge truly is (Cheng et al., 2024). Existing benchmarks also struggle to keep pace: once released, their contents may be absorbed into future training data, weakening their utility and leading to benchmark contamination. This phenomenon complicates the evaluation of retrieval-based methods, as models may have already memorized the relevant facts during training. In this paper, we propose that the solution to these challenges is **fast, automated, decentralized curation of dynamic knowledge benchmarks** that can track LLM development in real time and offer a clean testbed for evaluating retrieval augmented methods.

Despite the rapid advancement of LLMs and the growing need for accurate knowledge assessment, most popular benchmarks remain *static* after creation. Widely used datasets such as Natural Questions (Kwiatkowski et al., 2019), TriviaQA (Joshi et al., 2017), and HotpotQA (Yang et al., 2018) are primarily drawn from Wikipedia or curated text snapshots over a fixed time period. While instrumental in advancing open-domain question answering (QA) research, these benchmarks quickly become outdated and are often included in model pretraining corpora, leading to data contamination and inflated performance estimation (Li et al., 2024). More recent efforts such as StreamingQA (Liska et al., 2022), RealTimeQA (Kasai et al., 2023) and FreshQA (Vu et al., 2024) have begun including fresh facts. However, these dynamic benchmarks still rely on partial human curation and infrequent updates, or focus on a different task like forecasting. As a result, they don't enable continuous

---

[1]Code is available at `https://anonymous.4open.science/r/OKBench-0830`.

Table 1: Comparison of our benchmark with some previous knowledge QA & dynamic benchmarks in terms of objective, automation, update frequency, and scale.

| Benchmark | Objective | Automation | Update Freq. | Scale |
|---|---|---|---|---|
| StreamingQA | Factual QA | Partial | Static | $36, 800$ QA pairs |
| RealTime QA | Factual QA | Partial | Weekly | $\sim 30$ QA pairs |
| FreshQA | Factual QA & Debunking | Low | Weekly | 600 QA pairs, only update answers |
| LiveBench | Reasoning | Partial | Monthly | $40 - 100$ questions per task |
| Daily Oracle | Forecasting | Full | Daily | $\sim 17.2$ QA pairs |
| FutureX | Future Event Prediction | Full | Daily & Weekly | 500 events |
| Ours | Factual QA | Full | Any time | $\sim 2000$ QA pairs |

evaluation of LLMs' capabilities. Finally, these previous efforts are *centralized*, making it difficult and expensive to reproduce them on demand.

We propose an approach that addresses these challenges and democratizes dynamic knowledge benchmarking, by making it easy and practical for anyone to generate a new reproducible benchmark anytime. Specifically, we introduce **OKBench** (**O**pen**K**nowledge**B**ench), a fully automated framework for generating knowledge benchmarks for fair LLM evaluation. Focusing on the news domain where new knowledge emerges daily, our system automates the entire pipeline from information extraction to benchmark construction, producing multiple-choice QA items (with optional open-ended variants). **OKBench** is an agentic framework built on state-of-the-art LLMs, in which specialized agents for QA generation and validation collaborate to promote quality and consistency.

To enable benchmark generation at any time, we introduce a distribution and version control protocol that assigns each benchmark a unique signature, assuring consistent tracking and fair comparison across models and evaluations. The framework is fully *open-source*, empowering *any user* to generate up-to-date benchmarks *at any time*. This enables diverse use cases such as monitoring LLM knowledge freshness or evaluating retrieval-augmented models on clean, non-memorized data.

To assess the quality of the automatically generated benchmarks, we conduct manual validation of one of our question sets and find it to be of high quality.

To demonstrate the utility of our framework and assess current model capabilities, we evaluate a range of open-source and proprietary LLMs across multiple model sizes, with and without retrieval augmentation, using several retrieval strategies. Our results show a predictably large drop in performance when models are tested on new knowledge. Interestingly, when retrieval is introduced, the performance gap between smaller and larger models narrows significantly on knowledge not seen during training.

## 2  RELATED WORK

**Evaluating Retrieval-Augmented Generation (RAG)**  Retrieval-augmented generation (RAG) is a key strategy to equip large language models (LLMs) with up-to-date information by retrieving relevant external documents at inference time. However, existing RAG evaluations are often undermined by *data contamination*, where evaluation examples overlap with the model's pretraining corpus. This allows models to answer without retrieval, simply relying on memorized content (Li et al., 2024). Prominent QA datasets such as Natural Questions (Kwiatkowski et al., 2019), TriviaQA (Joshi et al., 2017), and HotpotQA (Yang et al., 2018) are sourced from Wikipedia or the open web, making it likely that models already "know" the answers. This undermines robust assessment of retrieval: models can appear strong simply by regurgitating seen content, and including training examples in prompts can further inflate performance (Wang et al., 2022). As a result, current benchmarks struggle to test whether models can truly retrieve and reason over novel information.

**Dynamic Knowledge Question Answering**  To address the limitations of static benchmarks, recent work has introduced *dynamic* QA benchmarks that reflect the evolving state of world knowledge.[2] StreamingQA (Liska et al., 2022) organizes questions chronologically over years of news data, but it does not support continual updates. RealTime QA (Kasai et al., 2023) delivers weekly quizzes

---

[2]For detailed descriptions of each benchmark, see Appendix A.

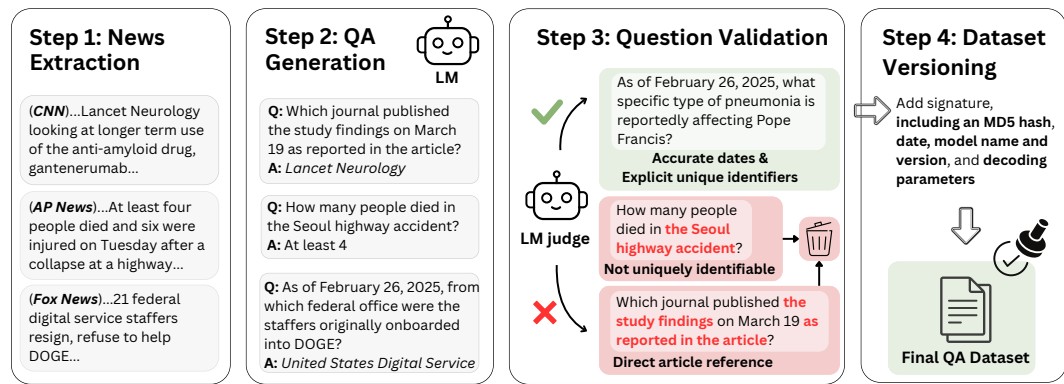

Figure 1: Automated dynamic knowledge benchmark construction pipeline for **OKBench**.

sourced from current news headlines, though coverage is limited by its limited breadth of news feeds and manual curation. FreshQA (Vu et al., 2024) is a centrally maintained benchmark of roughly 600 author- and freelancer-written, time-sensitive questions whose answers are periodically updated through extensive human annotation, making its ongoing maintenance costly and dependent on a single coordinating team.

**Dynamic Benchmarking Beyond QA**   Dynamic evaluation extends beyond question-answering to other formats. LiveCodeBench (Jain et al., 2025) harvests fresh programming-contest tasks to build a contamination-free, time-stamped suite for code generation, self-repair, execution, and test prediction; however, public releases so far reach only 2024-06-01 and still rely on semi-manual update scripts. LiveBench (White et al., 2025) is a challenging, contamination-limited LLM benchmark across multiple reasoning domains and offers partial updates monthly to guarantee a fresh suite of questions bi-yearly. AntiLeakBench (Wu et al., 2025) focuses on preventing data contamination by automatically constructing benchmarks sourced from Wikipedia, but it is constrained to Wikipedia updates. DeepScholar-Bench (Patel et al., 2025) focuses on automatic evaluation for generative research synthesis. Besides factual evaluation on recent knowledge, automatic benchmarks like Daily Oracle (Dai et al., 2025), ForecastBench (Karger et al., 2025) and FutureX (Zeng et al., 2025) test models on the task of forecasting near-future events. However, forecasting tasks aren't suitable for evaluating retrieval-based methods, as there's often no ground truth database to retrieve from. Therefore our focus is factual knowledge from the recent past.

Table 1 compares **OKBench** to other benchmarks along several dimensions. In summary, existing dynamic knowledge benchmarks still involve at least partial human curation, infrequent updates, or a narrow focus, and none offer a fully automated, large-scale solution for real-time factual knowledge evaluation. To our knowledge, **OKBench** is the first fully automated benchmark for evaluation of factual question answering ability.

## 3   AUTOMATED DYNAMIC BENCHMARKING WITH **OKBENCH**

### 3.1   BENCHMARK CONSTRUCTION PIPELINE

We design an agentic framework for dynamic knowledge benchmarking. The pipeline consists of four steps: (1) News extraction, (2) QA generation, (3) question validation, and (4) dataset versioning. An overview of the pipeline is shown in Figure 1.

**News Extraction**   We collect and preprocess news articles published within the past 24 hours from a diverse set of outlets, including both mainstream and specialized publications. The categorization and considered sources of news are presented in Table 2. Articles are retrieved via RSS feeds and parsed. For each article, we retain a structured representation that includes metadata such as the title, publication date, author, content body, and source URL. The output of this step is a curated, timestamped feed of news articles, which serves as the raw knowledge base for dynamic benchmark construction in subsequent stages.

Table 2: News sources used for dynamic knowledge extraction.

| Category | Sources |
|---|---|
| General / Mainstream News | CNN, BBC, Reuters, The Guardian, Fox News, NBC News, USA Today, HuffPost, CBS News |
| International Coverage | Al Jazeera, DW, RT, Channel News Asia (CNA), Times of India, South China Morning Post (SCMP) |
| Political Focus | Politico, The Hill, NPR |
| Technology and Science | TechCrunch, The Verge, Engadget, Ars Technica, Gizmodo, PC Gamer, TechRadar |
| Business / Finance | Bloomberg |
| Lifestyle / Culture | GQ, Vanity Fair |
| Open-Source Community News | WikiNews |

**QA Generation**  We use an LLM-based agent to generate initial multiple-choice question–answer pairs from curated news articles. The final questions can be delivered in either multiple-choice or open-ended format. The agent is instantiated using an LLM[3] guided by a prompt designed to elicit high-quality, time-sensitive questions (see Section B). The generation process involves identifying salient facts from each article, drafting a corresponding question, and producing one correct answer along with plausible distractor options. The agent is instructed to prioritize recent and unique facts, particularly entities, events, and developments that are unlikely to appear in older training data.

**Question Validation**  Despite detailed prompting, LLM-generated questions may not always be well suited for reliable model evaluation. In particular, the question sometimes explicitly refers to "the article," which is undesirable because we want every question to stand alone.

To address this, we introduce a dedicated question validation agent (see validation prompt in Section B) that assesses the quality and clarity of each question. The agent is tasked with verifying whether each question can be answered uniquely and unambiguously.

Specifically, it checks whether the question: (1) avoids direct references to the source article, (2) includes accurate and clear date references, (3) uses explicit identifiers for entities such as people, organizations, or events, and (4) avoids vague or ambiguous phrasing. Questions that fail any of these criteria are automatically discarded. Some example QA pairs created by our pipeline are shown in Table 3.

**Dataset Versioning**  To support reproducibility and fair comparison, each benchmark release is assigned a unique *signature* serving as its version identifier. Because dataset content can shift due to changes in daily news and the inherent stochasticity of LLM generation, we adopt a versioning approach inspired by SacreBLEU's reproducibility framework (Post, 2018). Each signature encodes the agent LLM model name and version (e.g., "gpt-4.1-2025-04-14"), the decoding hyperparameters (temperature, top-$p$, etc.), the dataset generation date and timestamp, and a randomly generated hash (specifically, MD5) as a unique identifier.

Users reporting results on our benchmarks should explicitly cite the full dataset signature and share the corresponding dataset snapshot. This enables precise reproduction and fair evaluation by others. By versioning each dataset and requiring explicit references, future work can reliably evaluate on the same benchmark instance, which is an essential safeguard in our decentralized benchmarking protocol, since numerous independently generated datasets may potentially exist.

## 3.2 HUMAN VALIDATION

To evaluate the quality of the generated QA pairs, we ran a human validation study on a set of pipeline outputs with two evaluation aspects:

• **Question Quality Check:** Does the question meet the clarity and unambiguity criteria?

• **Answer Correctness Check:** Does the provided correct option exactly match the source article?

Two independent panels of four computer-science PhD students (all native- or near-native English speakers) carried out the evaluations. The first panel is for evaluating question quality, where each evaluator assessed 60 multiple-choice questions. To better control for agreement, 20 questions in each annotator's part were simultaneously evaluated by 2 other annotators, each person 10 questions.

---

[3]We use gpt-4.1-2025-04-14 in our pipeline.

Table 3: Example generated QA pairs. The date of dataset generation is February 26, 2025.

| Question | Choices | Ground Truth |
|---|---|---|
| As of February 26, 2025, what percentage of GDP has UK Prime Minister Keir Starmer announced the country will spend on defense? | A. 2.3% of its GDP
B. 3% of its GDP
C. 2.5% of its GDP
D. 7% of its GDP | C. 2.5% of its GDP |
| On February 14, 2025, at which hospital was Pope Francis hospitalized for a respiratory infection? | A. St. Peter's Hospital
B. Vatican Medical Center
C. Gemelli Hospital
D. Apostolic Palace Clinic | C. Gemelli Hospital |
| In which year did Pope Francis have a piece of one lung removed? | A. 1967
B. 1955
C. 1947
D. 1957 | D. 1957 |
| On February 26, 2025, which individual from the Department of Psychiatry at the University of Cambridge emphasized the urgent need for new dementia treatments? | A. Dr. Marc Siegel
B. Dr. Ben Underwood
C. Dr. Chris Vercammen
D. Melissa Rudy | B. Dr. Ben Underwood |
| As of March 22, 2025, which journal published the study findings on March 19 that detailed the impact of gantenerumab on delaying Alzheimer's symptoms? | A. The Lancet Psychiatry
B. JAMA Neurology
C. Neurology
D. The Lancet Neurology | D. The Lancet Neurology |

Table 4: Example questions that do not pass the human evaluation.

| Question | Choices | Rationale |
|---|---|---|
| On what date was Pope Francis admitted to the hospital as of March 22, 2025? | A. Feb. 7
B. Feb. 14
C. Feb. 15
D. March 22 | Ambiguity; might rely on past information |
| When was George Foreman born? | A. September 24, 2011
B. October 30, 1974
C. January 10, 1949
D. June 3, 2009 | Rely on past information |
| Which streaming service is associated with Severance and previously known for hosting Ted Lasso, as of March 21, 2025? | A. Amazon Prime Video
B. Apple TV+
C. Hulu
D. Netflix | Rely on past information |
| As of March 22, 2025, updated HPV shots protect against how many strains of the virus? | A. nine
B. two
C. seven
D. eleven | Ambiguity and might rely on past information |

Based on question clarity, the average correctness rate is **92%** over 4 annotators on 200 questions in total. Table 4 shows some QA pairs that did not pass human evaluation. In the second panel, each evaluator independently rated answer correctness out of 25 QA pairs, and achieved **100%** correctness in the sampled questions.

The complete annotation guidelines and survey interfaces used in this study are provided in Section D and Section E. The complete evaluation results are in Section F. Because we aim for fully automated, decentralized usage, a small level of noise is acceptable to maintain scalability, freshness, and real-time evaluation. We also release a daily version of all news collected, enabling on-demand dataset generation under evolving knowledge conditions. As proprietary LLMs change over time, we will do periodic audits and updates to maintain consistent quality. By keeping human validation separate from the core pipeline, our framework remains cost-effective and adaptive, while still supporting quality control when needed.

## 3.3 DATASET STATISTICS AND COST ESTIMATION

Our pipeline ingests news articles from the previous 24 hours and typically yields ∼2,000 multiple-choice questions per run. For example, the snapshot generated on 22 March 2025 contains 2,350 questions. The end-to-end expense is modest: generating 2,350 raw questions with GPT-4.1-2025-04-14

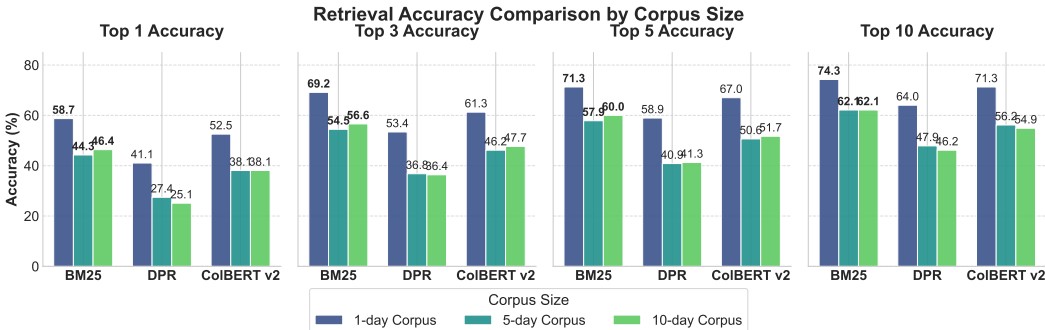

Figure 2: **Top-$k$ Retrieval Accuracy** for BM25, DPR, and ColBERT v2 across news corpora of different time windows (1-day, 5-day, and 10-day).

costs \$2.48. Validating the same 2,350 questions with the model costs an additional \$1.73. Consequently, a full daily benchmark costs roughly \$4.21, making on-demand generation practical for continuous evaluation.

## 3.4 QUESTION FORMATS

While the primary format of **OKBench** is multiple-choice questions, our pipeline can also generate an open-ended variant: for each article the generation agent poses a factoid question whose answer is a short span ($\leq 10$ tokens) copied verbatim from the article; during evaluation we let the model produce up to 100 tokens and pass its first non-empty line to a separate LLM judge, which simply checks string equality (after normalising case and punctuation) against the ground-truth span—that span being identical to the correct option in the MC version—and returns a binary correctness decision.

## 4 LLM EVALUATION EXPERIMENTS WITH **OKBENCH**

In the following experiments, we evaluate a set of models on the *March 22 snapshot* of the dataset (Section 3.3).[4] We evaluate a variety of open-source and proprietary LLMs. For the full list of models, please see Table 8.

**Evaluation Settings** We test each LLM under three information-access paradigms:

  (i) **No-Context**: The model sees only the question and answer choices. We simply provide the prompt: *"Question: {Q}. Provide the most accurate answer."* This reflects a purely parametric recall scenario, where the model must rely solely on its memorized knowledge.

 (ii) **Oracle-Context**: The model is given the ground-truth article (i.e., the document originally used to generate the question) as additional context. Here, the model input is of the form: *"Context: {Article}. Question: {Q}."*

(iii) **Retrieval**: We simulate a scenario where the model queries a recent news corpus and retrieves relevant articles before answering. We provide the top-$k$ passages (where $k \in \{1, 3, 5, 10\}$) returned by a retrieval system, concatenated into the prompt. The corpus is drawn from the last 24 hours (1-Day), the preceding 5 days (5-Day), or the preceding 10 days (10-Day). As the corpus grows, more outdated or irrelevant content is introduced, increasing retrieval difficulty.

---

[4]We focus on this single-day snapshot to provide a concrete, recent evaluation, though our framework can generate new benchmarks daily.

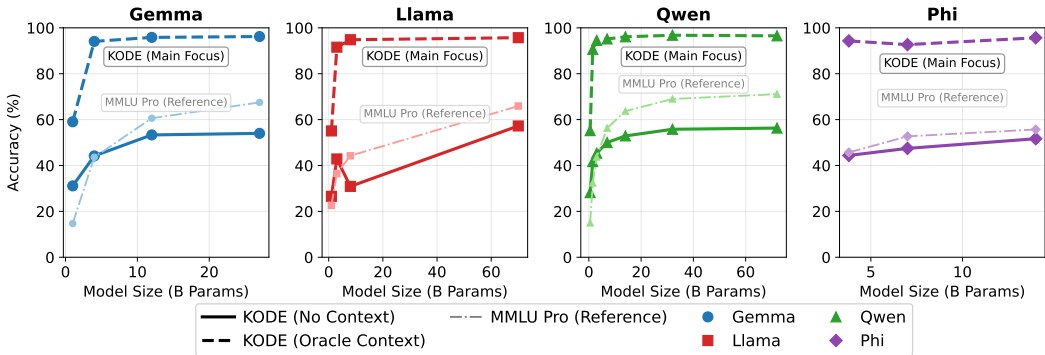

Figure 3: **No-Context vs. Oracle-Context QA Accuracy on OKBench**, plotted alongside each model's performance on MMLU Pro (lighter lines) as a reference for memorized knowledge. We show three representative model families (Gemma, Llama, Qwen) at various parameter scales (Billion Parameters). Solid lines denote *No-Context* accuracy (fresh knowledge), and dashed lines denote *Oracle-Context* accuracy when the ground-truth article is provided.

**Retrieval Methods**   We implement a variety of retrievers to provide context in the Retrieval setting. Each daily snapshot of news is indexed using **BM25 (lexical)** (Robertson & Zaragoza, 2009), a classic inverted-index-based method leveraging term frequency and inverse document frequency; **ColBERT v2 (dense)** (Santhanam et al., 2022), which encodes queries and documents at the token level and only then matches each query token to its most similar document token; and **DPR (dense)** (Karpukhin et al., 2020), a dual-encoder approach producing a single embedding per document and question, scored via dot product. For both dense retrievers, we use FAISS (Douze et al., 2025) with a flat index for approximate nearest neighbor search. We measure top-1, top-3, top-5, and top-10 retrieval accuracy (the fraction of queries where the ground-truth article is among the top-$k$ retrieved documents), as well as final QA performance after the model consumes those retrieved contents.

## 5 EVALUATION RESULTS AND DISCUSSION

### 5.1 LLM KNOWLEDGE VS. ORACLE-CONTEXT

Figure 3 summarizes the performance of four representative model families (Gemma, Llama, Qwen, Phi) on **OKBench** in both No-Context and Oracle-Context settings. Table 8 in Section G provides results for a more complete set of models. In addition to multiple-choice questions, we also report open-ended question-answering results in Section H. We find that the open-ended variant of the benchmark shows a trend similar to the multiple-choice variant, so we focus on the multiple-choice version here.

**Observation 1: Impact of Fresh Knowledge.**   When models must rely solely on parametric memory (No-Context), their performance is far from perfect across all sizes. This reflects the challenge of truly new facts that arise after the model's pretraining cutoff. Nevertheless, larger models do retain a slight edge. For instance, `gemma-3-1b-it` only achieves 31.1% accuracy in No-Context mode, whereas `gemma-3-27b-it` reaches 54.0%. The same trend appears in other families like Llama (26.6% vs. 57.2%) and Qwen (28.2% vs. 56.3%) when comparing the smallest and largest variants. In No-Context mode, Some "fresh-knowledge" questions still reference ongoing stories, such as an election that has been in the news for months, so even a small model can draw on background it has already seen and score above the 25% random-guess level. Bigger models possess an even richer store of that prior context, which is why they outperform the smaller ones despite the questions targeting very recent facts.

**Observation 2: Oracle-Context and a "Cutoff" for Reading Comprehension.**   Once the ground-truth article is appended to the query, most models (above a certain size threshold) quickly climb to high accuracy ($\sim 95\%$). Even a 4–7 B parameter model can answer correctly given the right passage, suggesting that *timely, precise* context is the main determinant of success. These findings underscore

that for fresh or real-time information, building robust retrieval pipelines may be more critical than simply scaling up model size.

However, contrary to the idea that *all* models do well with the article, Figure 3 and Table 8 in Section G show a sharp performance *cutoff*. Models around or above roughly 3–4 B parameters can read and understand the article sufficiently to push their Oracle accuracy to 90–95%. Yet *very small* LLMs (e.g., $\leq 1$ B parameters) achieve only around 55–60% even with the ground-truth article. This indicates a bound on reading comprehension capacity for extremely small models: they simply lack the representational power to parse the passage and correctly pinpoint the answer.

**Observation 3: Model Size Scaling behavior on Fresh Data vs. Memorized Knowledge.** The gap between smaller and larger models in the *No context* setting is smaller than one might expect from standard benchmarks that rely heavily on memorized knowledge. To illustrate this point, we also measured each model's performance on **MMLU Pro**, a knowledge-intensive benchmark widely used for assessing factual recall from pretraining. Figure 3 and Table 9 (in Section G.1) show that on MMLU Pro, scaling from a 1B to a 27B (or 70B) model often yields improvements of 40–50% or more; in contrast, for our newly generated QA data, the improvement over the same size range is 20–25%. Therefore, while model scale is critical for memorizing facts during pretraining, its benefits are more limited for *emergent* knowledge.

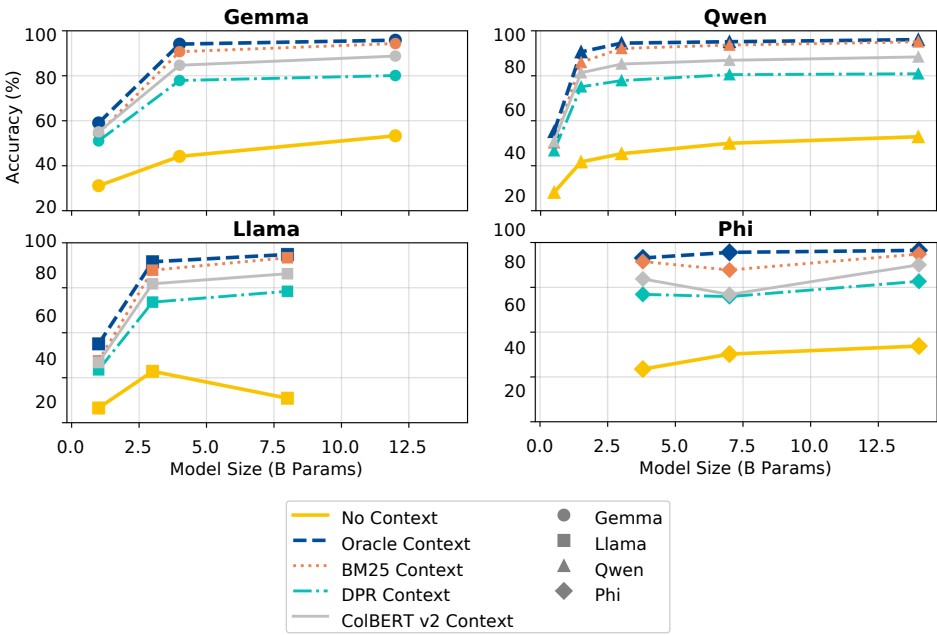

Figure 4: **QA Accuracy with Retrieval-Augmented Context.** Each panel shows QA accuracy (%) for four model families (Gemma, Llama, Qwen, Phi) across different parameter sizes, evaluated on dynamically generated news questions. Lines indicate performance with context retrieved by BM25 (dashed red), ColBERT v2 (solid gray), DPR (dash-dotted cyan), the Oracle (dashed blue, upper bound), and No-Context (solid yellow, lower bound) using top-3 retrieved passages from the 1-day news corpus.

## 5.2 RETRIEVAL PERFORMANCE

We experiment with three retrievers: **BM25**, **DPR**, and **ColBERT v2**. **Figure 2** shows their top-$k$ accuracy on daily news, while more detailed numerical results (e.g., top-1, top-3, etc.) are presented in **Appendix I** (Tables 11 and 12). Overall, BM25 achieves the highest top-$k$ accuracy in most settings, outperforming both DPR and ColBERT v2.

Interestingly, even though dense retrievers like DPR and ColBERT v2 often excel on standard benchmarks (Bajaj et al., 2018; Thakur et al., 2021), BM25 proves more robust for this dynamic

Table 5: Final QA accuracy (%) of LLMs under Retrieval settings, using `Llama-3.1-8B-Instruct` as the QA backbone. Retrieval is performed over 1-day, 5-day, and 10-day news corpora, returning top-$k$ passages ($k \in \{1, 3, 5, 10\}$).

| Retriever | 1-Day Corpus | | | | 5-Day Corpus | | | | 10-Day Corpus | | | |
|---|---|---|---|---|---|---|---|---|---|---|---|---|
| | Top-1 | Top-3 | Top-5 | Top-10 | Top-1 | Top-3 | Top-5 | Top-10 | Top-1 | Top-3 | Top-5 | Top-10 |
| BM25 | 90.47 | 93.49 | 93.40 | 92.60 | 88.43 | 91.79 | 92.89 | 92.04 | 88.30 | 91.15 | 92.26 | 92.09 |
| DPR | 66.26 | 77.66 | 81.28 | 84.21 | 59.49 | 70.89 | 74.34 | 78.13 | 57.53 | 68.60 | 71.57 | 75.96 |
| ColBERT v2 | 80.09 | 86.13 | 87.79 | 89.32 | 74.17 | 82.55 | 85.02 | 86.43 | 73.06 | 80.72 | 83.49 | 85.45 |

news scenario. The strong lexical cues (e.g., named entities, event-specific phrasing) may favor exact term matching. It also suggests that domain shift can hurt dense matching unless the models are further adapted, as they are typically trained on MS MARCO (Bajaj et al., 2018) (for ColBERT v2) or Natural Questions (Kwiatkowski et al., 2019)(for DPR) rather than on this news domain.

### 5.3 FINAL QA ACCURACY WITH RETRIEVED PASSAGES

Finally, we measure how these retrieval methods impact final question answering performance. Figure 4 shows the results using the 1-day news corpus and appending the top 3 retrieved documents across different models (full results in Section J). In line with the earlier retrieval results (cf. Figure 2), BM25-based retrieval also yields the highest end-to-end QA performance.

We also feed the top-$k$ passages from each retriever (BM25, DPR, ColBERT v2) into a moderate-scale `Llama-3.1-8B-Instruct` model and evaluate its QA accuracy. The complete results, including the final QA accuracy (%) across three corpus sizes (1-day, 5-day, and 10-day) and various $k$ values, are presented in Table 5. Overall, these results confirm that *accurate retrieval* is vital for time-sensitive QA, perhaps even more so than having a very large model. Even a 1.5B-parameter Qwen model achieves high QA accuracy (above 90%). Thus, for ever-evolving knowledge, robust retrieval pipelines can often compensate for a model's limited parametric memory.

## 6 CONCLUSION

We introduce a fully automated framework for dynamic knowledge benchmarking, enabling timely and decentralized evaluation of LLMs. Our agentic pipeline generates high-quality, news-driven QA datasets, supporting robust analysis of model knowledge and retrieval performance. Through experiments on a range of open-source and proprietary models, we demonstrate performance disparities on newly introduced knowledge and the benefits of retrieval augmentation. This work highlights the importance of evaluating LLMs on evolving, non-memorized knowledge to better understand and improve their real-world capabilities.

### STATEMENT ON LLM USAGE

We acknowledge the use of Large Language Models (LLMs) to assist in the preparation of this manuscript. Specifically, LLMs were utilized to improve grammar and clarity, aid in literature discovery, and generate boilerplate code snippets for our experiments and testing scripts. The authors have carefully reviewed and edited all LLM-generated outputs and take full responsibility for the final content and scientific integrity of this work.

### LIMITATIONS

While our framework democratizes the creation of *dynamic* knowledge benchmarks, several caveats remain:

- **Domain & Language Bias.** We currently target English-language online news. This excludes non-English, local, pay-walled, or multimedia sources and limits the benchmark's cultural and topical

coverage. Extending the pipeline to multilingual or domain-specific corpora (e.g., biomedical literature) will require tailored scraping, prompting, and validation strategies.

- **Dependence on Proprietary LLMs.** Generation and validation agents rely on proprietary frontier models. Model drift, API quota changes, or access restrictions may affect future reproducibility despite our version-signature protocol. Moreover, researchers without paid API access may face a cost barrier.

- **Legal and Ethical Considerations.** We scrape full-text news articles that remain under copyright. Our release distributes articles for research under fair-use assumptions, but downstream users bear responsibility for local licensing compliance. Automated harvesting also risks propagating misinformation if upstream outlets publish retracted or false content.

Addressing these limitations remains important future work for making dynamic knowledge evaluation truly global, robust, and sustainable.

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

## A ADDITIONAL BENCHMARK DETAILS

**StreamingQA.** Builds a time-indexed dataset from a large news corpus (14 years), enabling retrospective testing of how QA models adapt to new information at specific points in history. Once published, it is no longer updated.

**RealTime QA.** Scrapes around 30 weekly questions from news quizzes (e.g., CNN, The Week). Offers a rolling evaluation but is constrained by external quiz sources and weekly time slots, rather than daily updates.

**FreshQA.** Uses a fixed set of around 600 human-written questions whose answers evolve (often involving false premises or rapidly changing facts). Relies on regular human intervention for quality control and updating answers.

**Daily Oracle.** Automatically generates daily forecasting questions (T/F or multiple-choice) from current news, evaluating models' abilities to predict near-future outcomes. Fully automated, but does not focus on post-event factual retrieval or user-driven updates.

# B  PROMPT FOR GENERATING MCQS

```
# News article

**ARTICLE TITLE**:
{article_title}

**ARTICLE TEXT**:
{article_text}

**ARTICLE RELEASE DATE**:
{article_release_date}

# Your task

Generate 5 exceptionally challenging multiple-choice questions based on
    the article. Follow these requirements:

1. **Question Style**
   - Use a simple, direct tone. For example:
     - "Who was elected president of France in 2022?"
     - "Which country hosted the 2023 Climate Summit?"

2. **Question Content**
   - Each question must focus on factual information about the events or
       details within the article.
   - Formulate every question so it can be answered exclusively from the
       provided content.
   - Avoid referencing the article directly (do not use phrases like "
       According to the article..." or "The text indicates...").
   - For time-sensitive information, incorporate the article's release
       date. Use "as of {article_release_date}" when referring to ongoing
        or current information, or "on {article_release_date}" when
       indicating that an event occurred on that specific day.
   - Use explicit identifiers for individuals and organizations (e.g., "
       InfoWars reporter Jamie White"), never ambiguous references like "
       the official" or "his statement".
   - Ensure the question is only answerable if one has access to the
       article (low no-context accuracy).

3. **Answer Choices**
   - Provide four (4) plausible choices, each of which is the same entity
        type (person, organization, place, date, number, etc.).
   - The correct answer must be an entity present or derivable from the
       article.
   - Include distractors that are contextually plausible (either
       mentioned in the article or logically related).
   - At least one distractor should closely resemble the correct answer
       to increase difficulty (e.g., a similar name or date).
   - Use partial truths or common misconceptions for other distractors,
       ensuring all choices appear equally plausible without thorough
       reading.

4. **Answer Format**
   - Each question must have a single correct answer (entity) that is
       taken verbatim from the article.
   - The answer must not be open-ended: it should be a specific entity (
       person, organization, place, time, date, number, etc.).

5. **Question Diversity**
   - Cover different significant elements or events in the article (avoid
        repeating the same fact).
   - Use a variety of question types (who, what, when, where, why, how)
       and difficulty levels, from moderate to very challenging.
```

```
    - Aim to require different levels of reasoning (recall, inference,
        analysis).

6. **Article Release Date**  [IMPORTANT]
    - The article includes a release date provided as `{
        article_release_date}`. Ensure that this date is incorporated
        appropriately in questions, using "as of {article_release_date}"
        for current or ongoing contexts and "on {article_release_date}"
        when referencing a specific event or fact that happened that day.

7. **Response Format**
    - Return your final output as a JSON array of exactly 5 objects.
    - Each object must contain the following keys:
    - `"question_idx"`: An integer from 1 to 5.
    - `"question"`: A string containing the question text.
    - `"choices"`: An array of 4 strings, each a distinct answer option.
    - `"ground_truth"`: A string identical to the correct answer choice
        from `"choices"`.
    - `"rationale"`: A string explaining why the correct choice is
        correct and why the others are incorrect.

Now generate the JSON array with the specified structure:
```

## C  PROMPT FOR MCQ QUALITY CHECK

```
You are given a multiple-choice question in this format:

{qa_pair}

Check if it meets **all** of the following requirements:

1. **No direct reference to the article**
   - The question does not begin or contain phrases like "in the article
     ", "According to the article..." or "As reported in the article
     ...".

2. **Date references are accurate and clear**
   - If the question references an event or information that took place
     on a specific date, it can mention that date directly (e.g., "on
     February 25, 2025").
   - If the question references a continuing/ongoing situation relative
     to the article's publication, it should use "as of {
     article_release_date}" or "on {article_release_date}".
   - The question should not give ambiguous timing (e.g., "recently"
     without any date).

3. **Explicit identifiers for individuals or organizations**
   - Any person or group mentioned must be named clearly (e.g., "The
     Transportation Ministry" instead of "They" or "That ministry").
   - Avoid vague references like "the company" or "the government" if a
     specific entity is known.

4. **No ambiguous references**
   - If referencing a particular event, location, or study, the question
     must include all critical details known (e.g., event date,
     location, or official event name) so that it's clear which event
     or study is being discussed.
   - General phrases like "the collapse," "the incident," or "the study"
     are not acceptable. They must include identifying details such as
     the location, date, or name.

**Output "1" if *all* the requirements above are met, and "0" otherwise
  .**
```

# D HUMAN ANNOTATION GUIDELINES

## D.1 QUESTION QUALITY CHECK

We are evaluating the quality of a fresh knowledge benchmark dataset designed to test the latest information extracted from up-to-date news articles. This dataset consists of questions, multiple-choice answer candidates, and ground truth answers. Your task is to review the quality of this benchmarking data, specifically checking for clarity and freshness of questions, and the reasonableness of multiple-choice answers based on the provided news.

Please examine both the question and its multiple-choice options for major quality issues. Note that questions were generated on July 16, 2025, based on news articles published on that day.

Here are some guidelines on potential quality issues:

**Ambiguity**

The question itself should be clear and stand-alone. It should also have an unambiguous answer that is precisely one of the four choices. If you were given a set of articles containing the day's relevant news, you should be able to choose the correct answer. Consider the following:

- Questions should be answerable on their own. For example, phrasing like ". . . according to the article. . . " makes a question unclear.
- The time scope of the information requested should be clear. For current information, use phrasing such as "as of [Date]"; for past events on a given date, use "on [Date]."
- References to events, people, and entities should be clear. For example, use specific names (e.g., "Dr. Ben Underwood"), not vague references like "the official." However, titles or abbreviations are acceptable if they are unambiguous in context (e.g., "On July 16, 2025, the US President. . . ").

**Freshness**

We would like questions to be answerable using only the corresponding day's news, not previously known information. Examples of issues that diminish freshness include:

- Questions that rely on common sense or widely known facts, and can likely be answered without reading the source article.
- Historical trivia, such as questions with static answers (e.g., "What is Albert Einstein's birthday?" or "What was the US population in 2024?"), as these could be answered well before the question date of July 16, 2025.

**Question Quality Decision**

Question: [PLACE HOLDER]
Answer Candidate: [PLACE HOLDER]

Please indicate whether the question meets our quality criteria. If you are unsure, make your best guess and provide a comment explaining the uncertainty.

- Question passes the quality check
- Question is ambiguous/unanswerable
- Question is not fresh
- Question quality is not good for other reasons (please specify below)

**Any additional comments?**

Please use this space for any comments, such as if your question quality decision is uncertain, or if the question is of poor quality for unlisted reasons.

## D.2 QUESTION CORRECTNESS CHECK

**Human Evaluation Instructions**

You will work through 25 multiple-choice questions drawn from 5 different news articles.

The Google Form is organized into 15 sections—three sections for each article—so that you can follow the same three-step procedure every time.

**Three-step procedure (repeated for every article)**

1. Initial guess — prior knowledge only
   Read the question without looking at the article or any other source and choose the answer you think is correct.

2. Article-based answer
   Now read the accompanying news article carefully. Based solely on the information in the article, select the option that best answers the question.

3. Ground-truth check
   The ground-truth answer will be shown. Decide whether that answer is exactly supported by the article:
   Yes — it matches the article perfectly | No — it is contradicted or not stated.

**Form navigation**

1. Complete all three steps for the current article before clicking Next. Once you move to the next section you will not be able to return and edit earlier answers.

2. Repeat the three-step cycle until you have finished all 15 sections.

Thank you for taking the time to provide careful, accurate responses.

# E   HUMAN ANNOTATION INTERFACE

## E.1   QUESTION QUALITY CHECK

Figures 5 to 7 demonstrates the survey we used to collect human annotation results for quality checking our benchmark. We asked 4 annotators with various backgrounds and interests to each label 60 questions. Figure 5 is our general instructions which guides our annotators to label the 60 multiple-choice questions in Figure 6. We also collects each annotator's feedback after they finish evaluating all 60 questions to understand human concerns towards our benchmark.

## E.2   QUESTION CORRECTNESS CHECK

Figures 8 to 13 walk annotators through the three-step Google Form we use for the *Question Correctness Check*. Figure 8 presents the instructions page, which explains the task and navigation rules. In Step 1, shown in Figure 9, annotators make an *initial guess* for each multiple-choice question without reading the article. Step 2 begins with the full news article (Figure 10); after reading it, annotators answer the same questions again based solely on the article (Figure 11). Finally, Step 3 displays the ground-truth answers and asks annotators to judge whether they are exactly supported by the article (Figures 12 and 13).

## Open Knowledge Benchmark QA Quality Review

**General Instructions**

We are evaluating the quality of a fresh knowledge benchmark dataset designed to test the latest information extracted from up-to-date news articles. This dataset consists of questions, multiple-choice answer candidates, and ground truth answers. Your task is to review the quality of this benchmarking data, specifically checking for clarity and freshness of questions, and the reasonableness of multiple-choice answers based on the provided news. Please examine both the question and its multiple-choice options for major quality issues.

Note that questions were generated on **July 16, 2025**, based on news articles published on that day.

**Quality Guidelines for QA Human Eval**
Here are some guidelines on potential quality issues:
*1. Ambiguity:*

**The question itself should be clear and stand-alone. It should also have an unambiguous answer that is precisely one of the four choices.** If you were given a set of articles containing the day's relevant news, you should be able to choose the correct answer. Consider the following:

* **Questions should be answerable on their own.** For example, phrasing like "...according to the article..." makes a question unclear.
* **The time scope of the information requested should be clear.** For current information, use phrasing such as "as of [Date]"; for past events on a given date, use "on [Date]."
* **References to events, people, and entities should be clear.** For example, use specific names (e.g., "Dr. Ben Underwood"), not vague references like "the official." However, titles or abbreviations are acceptable if they are unambiguous in context (e.g., "On July 16, 2025, the US President...").

*2. Freshness:*

**We would like questions to be answerable using only the corresponding day's news, not previously known information**. Examples of issues that diminish freshness include:

* **Questions that rely on common sense or widely known facts**, and can likely be answered without reading the source article.
* **Historical trivia**, such as questions with static answers (e.g., "What is Albert Einstein's birthday?" or "What was the US population in 2024?"), as these could be answered well before the question date of July 16, 2025.

Thank you for taking time to complete our survey.

* Indicates required question

Figure 5: Instructions page of the Google Form used for the Question Quality Check. We ask our human annotators to assess clarity and freshness of the multiple-choice questions, based on the provided instructions.

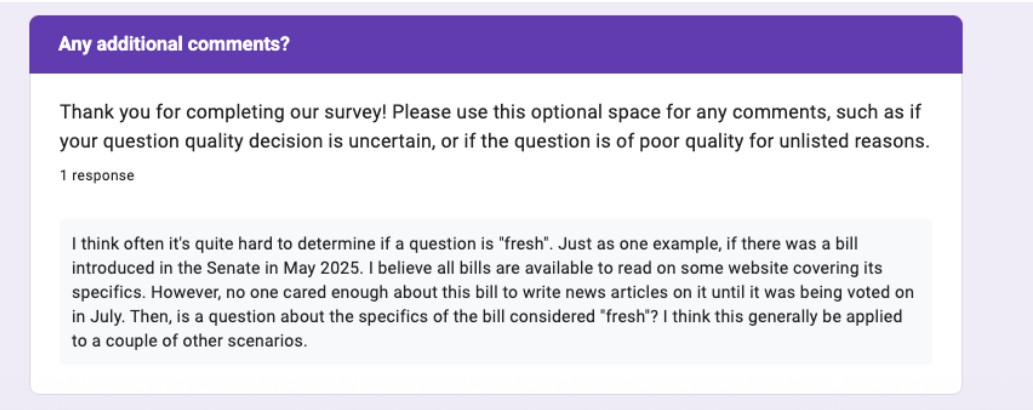

**Question Quality Decision**

Please indicate whether the question meets our quality criteria. If you are unsure, make your best guess and provide a comment explaining the uncertainty. There are 60 questions in total.

What is the primary reason given by Rep. Mark Harris for seeking to revoke the National Education Association's federal charter, as reported on July 16, 2025? *

A. The NEA opposed increased teacher salaries.
B. The NEA failed to support public charter schools.
C. The NEA mismanaged federal funds.
D. The NEA has become a partisan advocacy group.

○ Question passes the quality check

○ Question is ambiguous/unanswerable

○ Question is not fresh

○ Question quality is not good for other reasons (please specify below)

○ Other: ___________

Figure 6: One QA pair example taken from our Question Quality Check survey.

**Any additional comments?**

Thank you for completing our survey! Please use this optional space for any comments, such as if your question quality decision is uncertain, or if the question is of poor quality for unlisted reasons.

1 response

I think often it's quite hard to determine if a question is "fresh". Just as one example, if there was a bill introduced in the Senate in May 2025. I believe all bills are available to read on some website covering its specifics. However, no one cared enough about this bill to write news articles on it until it was being voted on in July. Then, is a question about the specifics of the bill considered "fresh"? I think this generally be applied to a couple of other scenarios.

Figure 7: Comments section in our Question Quality Check Survey.

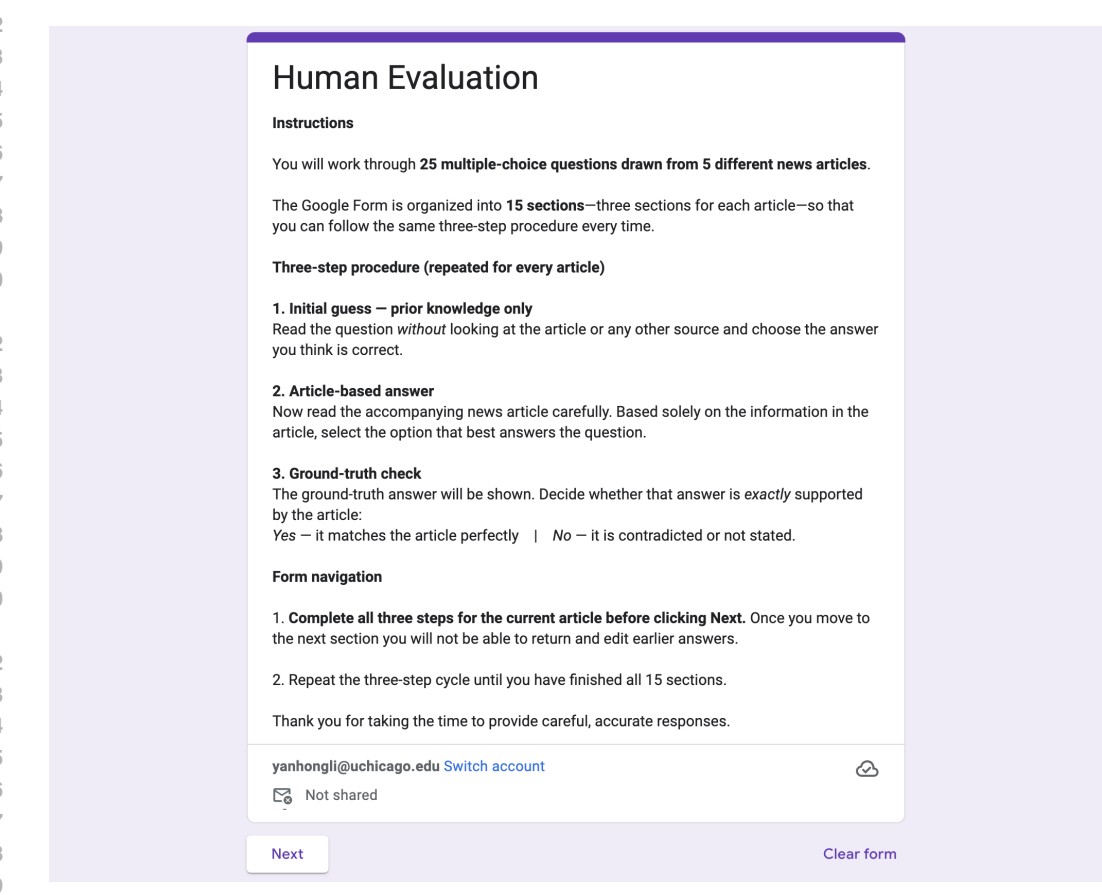

Figure 8: Instructions page of the Google Form used for the Question Correctness Check. It summarises the three-step workflow and navigation rules for annotators.

Figure 9: Step 1 (**Initial guess**). Annotators answer each multiple-choice question based only on prior knowledge, before seeing the article.

Figure 10: Step 2 (**Article reading**). The news article is presented in full so annotators can consult it before answering again.

grounds, shelling and firing on the compound. Sudan has faced years of chaos and war Sudan, a nation in northeastern Africa, has been unstable since a popular uprising forced the removal of longtime autocratic President Omar al-Bashir in 2019. A short-lived transition to democracy was derailed when Burhan and Dagalo led a military coup in 2021. The RSF and Sudan's military began fighting each other in 2023. Since the start of the year, Burhan's forces, including Sudan's military and allied militias, have advanced against the RSF. They retook a key refinery north of Khartoum, pushed in on RSF positions around Khartoum itself. The fighting has led to an increase in civilian casualties. Al-Bashir faces charges at the International Criminal Court over carrying out a genocidal campaign in the early 2000s in the western Darfur region with the Janjaweed militia, the RSF precursor. Rights groups and the U.N. accuse the RSF and allied Arab militias of again attacking ethnic African groups in this latest war. CLICK HERE TO GET THE FOX NEWS APP Since the war began, both the Sudanese military and the RSF have faced allegations of human rights abuses. Before U.S. President Joe Biden left office, the State Department declared the RSF are committing genocide. The military and the RSF have denied committing abuses.

Who is the army chief leading Sudan's military advances as of March 21, 2025? *

○ Gen. Mohammed Hamdan Dagalo

○ Gen. Omar al-Bashir

○ Gen. Abdel-Fattah Burhan

○ Gen. Khaled al-Aiser

○ I do not know

Which location did Sudan's military retake on March 21, 2025? *

○ Arab Market

○ Al-Maliha

○ Republican Palace

○ Khartoum International Airport

○ I do not know

Figure 11: Step 2 (**Article-based answer**). After reading the article, annotators choose the option that best answers each question.

## Human Evaluation

yanhongli@uchicago.edu  Switch account

☒ Not shared

* Indicates required question

### Section 3 (Article 1, Step 3)

The ground-truth answer will be shown. Decide whether that answer is *exactly* supported by the article:

*Yes* — it matches the article perfectly   |   *No* — it is contradicted or not stated.

**Sudan's military says it has retaken Khartoum's Republican Palace, seat of country's government**
Fox News Flash top headlines are here. Check out what's clicking on Foxnews.com. Sudan's military on Friday retook the Republican Palacein Khartoum, the last heavily guarded bastion of rival paramilitary forces in the capital, after nearly two years of fighting. The seizure of the Republican Palace, surrounded by government ministries, was a major symbolic victory for Sudan's military against the paramilitary Rapid Support Forces — though it likely doesn't mean the end of the war as the RSF holds territory in Sudan's western Darfur region and elsewhere. SUDAN'S ARMY DENOUNCES VIDEO ALLEGEDLY SHOWING ITS TROOPS CARRYING SEVERED HEADS OF ENEMIES Social media videos showed Sudanese soldiers inside the palace, giving the date as the 21st dayof Ramadan, the holy Muslim fasting month, which corresponds to Friday. A Sudanese military officer wearing a captain's epaulettes made the announcement in the video and confirmed the troops were inside the compound. The palace appeared to be in ruins, with soldiers' stepping on broken tiles. Troops carrying assault rifles and rocket-propelled grenade launchers chanted: "God is the greatest!" Khaled al-Aiser, Sudan's information minister, said the military had retaken the palace in a post on the social platform X. An army soldier walks in front of the Republican Palace in Khartoum, Sudan, after it was taken over by Sudan's army Friday, March 21, 2025. (AP Photo) "Today the flag is raised, the palace is back and the journey continues until victory is complete," he wrote. Later, curious residents wandered through the palace. Walls stood pockmarked by rifle rounds. Smears of blood led to dead bodies, covered haphazardly with blankets. Palace's fall a symbolic and strategic moment The fall of the Republican Palace — a compound along the Nile River that was the seat of government before the war erupted and is immortalized on Sudanese banknotes and postage stamps — marks another battlefield gain for Sudan's military, which has made steady advances in recent months under army chief Gen. Abdel-Fattah Burhan. It also means that the rival RSF fighters, under Gen. Mohammed Hamdan Dagalo, have been mostly expelled from the capital, Khartoum.

Figure 12: Step 3 (**Ground-truth verification instructions**). The form explains how to judge whether the provided ground-truth answer is exactly supported by the article.

malnourished children at Al Bashir Hospital on Khartoum's outskirts. "Commercial supplies and humanitarian aid have been blocked for more than three months due to ongoing conflict along key routes," UNICEF warned. "The result is a severe shortage of food, medicine and other essentials, with thousands of civilians trapped in active fighting." The war has killed more than 28,000 people, forced millions to flee their homes and left some families eating grass in a desperate attempt to survive as famine sweeps parts of the country. Other estimates suggest a far higher death toll. The Republican Palace became the seat of power during the British colonization of Sudan. It also saw some of the first flags of independent Sudan raised in 1956. The complex had also been the main office of Sudan's president and other top officials. The Sudanese military has long targeted the palace and its grounds, shelling and firing on the compound. Sudan has faced years of chaos and war Sudan, a nation in northeastern Africa, has been unstable since a popular uprising forced the removal of longtime autocratic President Omar al-Bashir in 2019. A short-lived transition to democracy was derailed when Burhan and Dagalo led a military coup in 2021. The RSF and Sudan's military began fighting each other in 2023. Since the start of the year, Burhan's forces, including Sudan's military and allied militias, have advanced against the RSF. They retook a key refinery north of Khartoum, pushed in on RSF positions around Khartoum itself. The fighting has led to an increase in civilian casualties. Al-Bashir faces charges at the International Criminal Court over carrying out a genocidal campaign in the early 2000s in the western Darfur region with the Janjaweed militia, the RSF precursor. Rights groups and the U.N. accuse the RSF and allied Arab militias of again attacking ethnic African groups in this latest war. CLICK HERE TO GET THE FOX NEWS APP Since the war began, both the Sudanese military and the RSF have faced allegations of human rights abuses. Before U.S. President Joe Biden left office, the State Department declared the RSF are committing genocide. The military and the RSF have denied committing abuses.

Who is the army chief leading Sudan's military advances as of March 21, 2025? *
**Ground Truth: Gen. Abdel-Fattah Burhan**

○ Yes

○ No

Which location did Sudan's military retake on March 21, 2025? *
**Ground Truth: Republican Palace**

○ Yes

○ No

Figure 13: Step 3 (**Ground-truth judgment**). Annotators indicate *Yes* if the ground-truth matches the article or *No* otherwise for each question.

Table 6: Human annotation results for the Question Quality Check. Each annotator is responsible for labeling 60 multiple-choice questions, where they assess question clarity and knowledge freshness. For each annotator's part, 20 out of 60 questions are annotated by 2 different annotators to calculate the agreement ratio. In total, 200 questions are evaluated for quality.

| Annotator | Question Range | Failed Clarity | Failed Freshness | Correctness | Agreement |
| --- | --- | --- | --- | --- | --- |
| | indices | # ambiguous questions | # outdated questions | # passed questions | # matched failures |
| 1 | 1-60 | 5 | 4 | 51 | 0/20 |
| 2 | 51-110 | 1 | 11 | 45 | 0/20 |
| 3 | 101-160 | 3 | 26 | 31 | 3/20 |
| 4 | 1-10, 151-200 | 10 | 17 | 31 | 1/20 |

Table 7: Human annotation results for the Question Correctness Check. Part 1 assesses prior-knowledge accuracy, Part 2 agreement with the multiple-choice ground truth after reading the article, and Part 3 verification that the ground-truth answer is fully supported by the article.

| Annotator | Part 1 | Part 2 | Part 3 |
| --- | --- | --- | --- |
| | # correct guesses | # matched answers | # "Yes" judgments |
| 1 | 1 | 25 | 25 |
| 2 | 1 | 25 | 25 |
| 3 | 2 | 25 | 13 |
| 4 | 6 | 25 | 25 |
| **Total (out of 100)** | 10 | 100 | 88 |

# F    HUMAN ANNOTATION RESULTS

## F.1    QUESTION QUALITY CHECK

Table 6 shows the human evaluation results of *Question Quality Check*. Across four annotators, the average correctness rate is 92% based on clarity only. It's noteworthy that our human annotators mostly disagree on freshness, since none of the questions was labeled as ambiguous by both annotators. Out of 40 overlapped questions, there are 2 questions labeled as not fresh by both annotators, and 1 question where both annotators labeled as fail out of different reasons. We observe that assessing question freshness is generally more difficult, due to the limitations of news articles, humans and LLMs. As noted by one of our annotators, it's challenging to determine if a question is "fresh" in multiple scenarios, since some seemingly trivial information from the past may be randomly covered by future news articles. Therefore, we consider clarity as the primary metric to assess the quality of our benchmark, as freshness can be easily affected by a lot of uncontrollable factors. When evaluated on both clarity and freshness, the average correctness rate is 66% on 200 multiple-choice questions.

## F.2    QUESTION CORRECTNESS CHECK

Table 7 summarises the outcomes of the three-step *Question Correctness Check*. Part 1 measures how often annotators could guess the correct answer *before* reading the article; the low 10 % accuracy confirms that the questions are not answerable from prior knowledge alone. After consulting the article (Part 2), all annotators selected the ground-truth multiple-choice option in every case (100% agreement), indicating the questions are clear and the correct choice is recoverable from the article. In Part 3, annotators judged whether the ground-truth answer is *exactly* supported by the article; 88 % of judgments were "Yes". All 12 disputed items were produced by the same annotator, who later acknowledged that they were unsure of the date of the new passage and therefore over-thought their answers. Therefore, the misunderstanding stems from the questionnaire design, not from the answers being incorrect.

# G    COMPLETE MODEL BENCHMARKING RESULTS (MULTIPLE CHOICE FORMAT)

Table 8 shows the final QA accuracy (%) for a broad range of open-sourced and proprietary LLMs under both *No-Context* and *Oracle* settings. As discussed in the main paper, these results highlight the importance of timely context for questions involving fresh, real-world information and illustrate a performance "cutoff" phenomenon for smaller model sizes (e.g., 1B parameters) versus larger ones (e.g., 7B or more). "Oracle" accuracy steadily approaches near-ceiling for models above roughly 3–4B parameters, indicating a scaling threshold for effective reading comprehension on time-sensitive content.

Table 8: Final QA accuracy (%) of open-sourced and closed-sourced LLMs under No-Context and Oracle (Context) settings.

| Model | No-Context Acc | Oracle Acc |
|---|---|---|
| **Open-Sourced Models** | | |
| gemma-3-1b-it | 31.11 | 59.06 |
| gemma-3-4b-it | 44.17 | 94.09 |
| gemma-3-12b-it | 53.32 | 95.83 |
| gemma-3-27b-it | 54.00 | 96.21 |
| Llama-3.2-1B-Instruct | 26.55 | 55.06 |
| Llama-3.2-3B-Instruct | 42.85 | 91.57 |
| Llama-3.1-8B-Instruct | 30.89 | 94.81 |
| Llama-3.3-70B-Instruct | 57.23 | 95.70 |
| Phi-3-mini-128k-instruct | 44.38 | 94.30 |
| Phi-3-small-128k-instruct | 47.45 | 92.68 |
| Phi-3-medium-128k-instruct | 51.66 | 95.66 |
| Phi-4-mini-instruct | 43.57 | 93.62 |
| Qwen2.5-0.5B-Instruct | 28.17 | 55.19 |
| Qwen2.5-1.5B-Instruct | 41.70 | 90.64 |
| Qwen2.5-3B-Instruct | 45.36 | 94.51 |
| Qwen2.5-7B-Instruct | 50.00 | 95.15 |
| Qwen2.5-14B-Instruct | 52.89 | 96.09 |
| Qwen2.5-32B-Instruct | 55.79 | 96.77 |
| Qwen2.5-72B-Instruct | 56.30 | 96.51 |
| Mistral-7B-Instruct-v0.2 | 35.96 | 90.21 |
| Mistral-Small-24B-Instruct-2501 | 53.23 | 96.43 |
| Mixtral-8x7B-Instruct-v0.1 | 33.36 | 93.40 |
| **Proprietary Models** | | |
| GPT-4o | 59.96 | 96.60 |
| GPT-o1-mini | 32.38 | 96.34 |
| GPT-o3-mini | 55.36 | 97.28 |
| Gemini-1.5-pro | 55.36 | 97.28 |

## G.1  MMLU PRO: MEMORIZED KNOWLEDGE ASSESSMENT

Table 9: **MMLU Pro Results** (% accuracy). We report performance on a knowledge-intensive QA benchmark, reflecting memorized or static knowledge from pre-training.

| Model | Size | Accuracy (%) |
|---|---|---|
| Llama-3.2-1B-Instruct | 1B | 22.6 |
| Llama-3.2-3B-Instruct | 3B | 36.5 |
| Llama-3.1-8B-Instruct | 8B | 44.3 |
| Llama-3.3-70B-Instruct | 70B | 65.9 |
| Gemma-3-1B | 1B | 14.7 |
| Gemma-3-4B | 4B | 43.6 |
| Gemma-3-12B | 12B | 60.6 |
| Gemma-3-27B | 27B | 67.5 |
| Qwen-2.5-0.5B | 0.5B | 15.0 |
| Qwen-2.5-1.5B | 1.5B | 32.4 |
| Qwen-2.5-3B | 3B | 43.7 |
| Qwen-2.5-7B | 7B | 56.3 |
| Qwen-2.5-14B | 14B | 63.7 |
| Qwen-2.5-32B | 32B | 69.0 |
| Qwen-2.5-72B | 72B | 71.1 |

In Table 9, we report the accuracy of various models on the MMLU Pro benchmark, a knowledge-intensive QA dataset aimed at evaluating factual recall from pre-training. These results offer insight into how well each model retains *static* domain knowledge, in contrast to the *dynamic*, newly emerging facts tested by our daily-updated QA benchmark. We observe that scaling model size often brings significant improvements in MMLU Pro accuracy, reflecting the growing capacity for memorizing factual content. Notably, the performance gains on MMLU Pro can be substantially larger than the gains observed on our fresh-news dataset under No-Context conditions, underscoring the difference between learned "long-term" knowledge and newly introduced facts.

## H    COMPLETE MODEL BENCHMARKING RESULTS (OPEN ENDED QUESTIONS)

Table 10 reports the open-ended question-answering accuracy (%) of every model we evaluated under both *No-Context* and *Oracle* settings. The table consolidates results for open-sourced and closed-sourced LLMs, making it easy to trace how providing the exact answer-containing passage ("Oracle") closes the gap that appears when models must rely solely on their parametric knowledge ("No-Context"). Reading downward, you can also see the scale threshold—around 3-4B parameters—beyond which Oracle accuracy plateaus near ceiling, while smaller models lag substantially without context.

Table 10: Final accuracy (%) of all tested models in the *No-Context* and *Oracle* settings usin open-ended question format.

| Model | No-Context Acc | Oracle Acc |
|---|---|---|
| **Open-Sourced Models** | | |
| gemma-3-1b-it | 4.64 | 70.09 |
| gemma-3-4b-it | 10.51 | 86.09 |
| gemma-3-12b-it | 14.17 | 89.79 |
| gemma-3-27b-it | 17.19 | 89.79 |
| Llama-3.2-1B-Instruct | 2.81 | 74.72 |
| Llama-3.2-3B-Instruct | 5.19 | 86.64 |
| Llama-3.1-8B-Instruct | 2.68 | 82.94 |
| Llama-3.3-70B-Instruct | 16.13 | 90.13 |
| Phi-3-mini-128k-instruct | 8.26 | 82.64 |
| Phi-4-mini-instruct | 8.98 | 75.19 |
| Qwen2.5-0.5B-Instruct | 4.47 | 70.94 |
| Qwen2.5-1.5B-Instruct | 7.02 | 85.53 |
| Qwen2.5-3B-Instruct | 6.98 | 86.85 |
| Qwen2.5-7B-Instruct | 9.36 | 89.62 |
| Qwen2.5-14B-Instruct | 10.43 | 84.68 |
| Qwen2.5-32B-Instruct | 11.36 | 91.02 |
| Qwen2.5-72B-Instruct | 13.79 | 90.94 |
| Mistral-7B-Instruct-v0.2 | 5.19 | 84.47 |
| Mistral-Small-24B-Instruct-2501 | 12.77 | 90.64 |
| Mixtral-8x7B-Instruct-v0.1 | 8.60 | 86.34 |
| **Closed-Sourced Models** | | |
| GPT-4o-2024-08-06 | 17.74 | 91.79 |
| o1-mini-2024-09-12 | 4.43 | 88.30 |
| o3-mini-2025-01-31 | 15.79 | 92.38 |
| Gemini-1.5-pro | 17.83 | 90.13 |

# I  ADDITIONAL RETRIEVAL RESULTS

To provide a fuller picture of how our retriever stack behaves under different temporal scopes, Table 11 details the Top-$k$ hit rates—the fraction of questions whose gold article appears within the first $k$ results—while Table 12 complements this view with Top-$k$ mean reciprocal rank (MRR), capturing average ranking quality. We report both metrics for BM25, DPR, and ColBERT v2 across three corpus sizes (news from the last 1, 5, and 10 days) and four cut-off values ($k = 1, 3, 5, 10$). Together, these tables reveal how retrieval effectiveness degrades as the candidate pool widens, and how each method trades off early-precision (Top-1/3) versus broader recall (Top-10) under increasingly challenging settings.

Table 11: Top-$k$ hits accuracy (%) for different retrieval methods across 1-day, 5-day, and 10-day corpora. Each cell represents the fraction of questions for which the ground-truth article is ranked within the top $k$ results.

| Retriever | 1-Day Corpus | | | | 5-Day Corpus | | | | 10-Day Corpus | | | |
|---|---|---|---|---|---|---|---|---|---|---|---|---|
| | Top-1 | Top-3 | Top-5 | Top-10 | Top-1 | Top-3 | Top-5 | Top-10 | Top-1 | Top-3 | Top-5 | Top-10 |
| BM25 | 58.72 | 69.15 | 71.28 | 74.26 | 44.26 | 54.47 | 57.87 | 62.13 | 46.38 | 56.60 | 60.00 | 62.13 |
| DPR | 41.06 | 53.40 | 58.94 | 64.04 | 27.45 | 36.81 | 40.85 | 47.87 | 25.11 | 36.38 | 41.28 | 46.17 |
| ColBERT v2 | 52.55 | 61.28 | 67.02 | 71.28 | 38.09 | 46.17 | 50.64 | 56.17 | 38.09 | 47.66 | 51.70 | 54.89 |

Table 12: Top-$k$ Mean Reciprocal Rank (MRR) for different retrieval methods across 1-day, 5-day, and 10-day corpora. Each cell represents the average reciprocal rank of the ground-truth article.

| Retriever | 1-Day Corpus | | | | 5-Day Corpus | | | | 10-Day Corpus | | | |
|---|---|---|---|---|---|---|---|---|---|---|---|---|
| | Top-1 | Top-3 | Top-5 | Top-10 | Top-1 | Top-3 | Top-5 | Top-10 | Top-1 | Top-3 | Top-5 | Top-10 |
| BM25 | 0.59 | 0.63 | 0.64 | 0.64 | 0.44 | 0.49 | 0.50 | 0.50 | 0.46 | 0.51 | 0.52 | 0.52 |
| DPR | 0.41 | 0.47 | 0.48 | 0.49 | 0.27 | 0.32 | 0.32 | 0.33 | 0.25 | 0.30 | 0.31 | 0.32 |
| ColBERT v2 | 0.53 | 0.56 | 0.58 | 0.58 | 0.38 | 0.42 | 0.43 | 0.43 | 0.38 | 0.43 | 0.43 | 0.44 |

Table 13: End-to-end QA accuracy (%) on the 1-day news corpus with the top-3 retrieved passages appended to each query.

| Model | BM25 | DPR | ColBERT v2 |
|---|---|---|---|
| Gemma-3-1B-IT | 54.43 | 51.06 | 55.06 |
| Gemma-3-4B-IT | 90.72 | 77.91 | 84.68 |
| Gemma-3-12B-IT | 94.34 | 80.11 | 88.77 |
| Gemma-3-27B-IT | 95.28 | 77.19 | 86.77 |
| Llama-3.2-1B-Instruct | 47.49 | 43.53 | 46.77 |
| Llama-3.2-3B-Instruct | 87.83 | 73.62 | 81.79 |
| Llama-3.1-8B-Instruct | 93.36 | 78.43 | 86.26 |
| Llama-3.3-70B-Instruct | 94.98 | 78.13 | 86.98 |
| Qwen 2.5-0.5B-Instruct | 50.17 | 46.68 | 50.77 |
| Qwen 2.5-1.5B-Instruct | 85.96 | 75.11 | 81.36 |
| Qwen 2.5-3B-Instruct | 92.17 | 77.87 | 85.23 |
| Qwen 2.5-7B-Instruct | 93.66 | 80.51 | 86.89 |
| Qwen 2.5-14B-Instruct | 95.11 | 80.89 | 88.38 |
| Qwen 2.5-32B-Instruct | 96.00 | 84.21 | 89.96 |
| Qwen 2.5-72B-Instruct | 95.45 | 85.02 | 90.43 |
| Phi-3-mini-128k-Instruct | 91.49 | 76.85 | 83.74 |
| Phi-4-mini-Instruct | 91.79 | 78.55 | 83.45 |
| Phi-3-small-128k-Instruct | 87.74 | 75.87 | 76.81 |
| Phi-3-medium-128k-Instruct | 94.85 | 82.68 | 90.04 |

## J COMPLETE END-TO-END QA RESULTS

Table 5 reports the final question-answering accuracy (%) when each LM receives the top 3 passages returned by three retrievers—BM25, DPR, and ColBERT v2—on the 1-day news corpus. These numbers complement Figure 4 by revealing how retrieval quality interacts with model size *across the full model set*. Higher accuracies for BM25 corroborate our main-text claim that lexical cues (named entities, dates) dominate in rapidly evolving news, while dense retrievers lag unless adapted to the domain.

