# OpenReview forum: "OKBench: Democratizing LLM Evaluation with Fully Automated, On-Demand Open Knowledge Benchmarking"
_ICLR.cc/2026/Conference — Submitted to ICLR 2026_

### Official Review · Reviewer_hfc4 · 2025-10-27

**Soundness:** 2
**Presentation:** 3
**Contribution:** 2
**Rating:** 4
**Confidence:** 3

**Summary:**

This paper presents an automatic benchmark construction method for evaluating LLMs with multi-choice questions based on recent news that are expected to be not memorized by LLMs. The construction method uses LLM to generate questions and validate questions. 4 PhD students are asked to validate 200 questions with two panels: the answers have a correct rate of 94% in the first panel and 100% in the second panel.

The evaluate reports results of testing four LLMs, without context, without oracle context and with retrievers of BM25, DPR and ColBERT.

**Strengths:**

1. Evaluating the performance of LLMs and LLMs with RAG is an interesting research problem. The dataset attempts to bridge this gap.

2. The paper is well written.

3. A lot of results on different LLMs, with and without context, and different retrievers are reported and analysed.

**Weaknesses:**

1. The current evaluation focuses on different LLM-based QA models, but ignore the dataset construction method, and the key features of the dataset.

2. I do not see any results on the efficiency and scalability of the benchmark construction method. High efficiency is required to ensure emerging LLMs. The validation of the question-answer quality depends on human annotation. This is time consuming. I also think this manual validation is required when new samples are generated, as the effectiveness of the LLMs for creating these samples may expire in dealing with new data.

3. There is a shortage of incremental evaluation, e.g., results of datasets constructed in Jan 2025, Apr 2025, Jul 2025. This can observe the effectiveness of the data construction method in the time line.

**Questions:**

1. As the questions are new (after the LLM is pre-trained), why some LLMs can still achieve as good performance as 50%?

2. Is the following paper, which use new knowledge from Wikidata for evaluating LLMs, relevant? https://arxiv.org/abs/2412.17032

3. How do you check the licenses or copyright of the news data providers? Does the news data include any sensitive personal information?

**Details Of Ethics Concerns:**

We need to check whether the authors satisfy the copyright of the news data providers, and whether the news data involve any personal information.

---

> ### Author Response · Authors · 2025-12-03
>
> Dear reviewer hfc4,
>
> We appreciate the chance to address each of your concerns below:
>
> "1. The current evaluation focuses on different LLM-based QA models, but ignore the dataset construction method, and the key features of the dataset."
>
> We want to clarify that the goal behind OKBench is to evaluate whether language models can adapt to fresh knowledge with retrieval augmentation, which is the reason we invested substantial effort on evaluating several families of language models. With that said, we believe that data construction and the key features of our benchmark are not ignored. We have a detailed description of the agentic pipeline, quality control and benchmark versioning in Section 3.1. We also provide key features of the dataset (source, size, examples, deeper analysis such as human qualitative evaluation) in Section 3.2, 3.3, 3.4 and Table 2, 3 & 4.  We are not sure what additional details the reviewer would have liked to see.
>
> "2. I do not see any results on the efficiency and scalability of the benchmark construction method. High efficiency is required to ensure emerging LLMs. The validation of the question-answer quality depends on human annotation. This is time consuming. I also think this manual validation is required when new samples are generated, as the effectiveness of the LLMs for creating these samples may expire in dealing with new data."
>
> Our human annotation results are there just to demonstrate the quality of our benchmark. In real-world use cases, we don't rely on manual validation to generate new samples. We agree that the effectiveness of LLMs may expire, but it's a drawback of every agentic pipeline. We designed specific mechanisms to control the quality of generated data, where an iterative generator-validator framework with automatic LLM judgements is adopted. More computation with multi-round iterations canfurther improve  quality. The effectiveness of the pipeline in terms of data quality is verified by humans in our study (Section 3.2 and Appendix D). To clarify, we do not require human evaluation to accompany every version of benchmark generation. It is simply an option that we demonstrate to probe the data quality periodically.
>
> "3. There is a shortage of incremental evaluation, e.g., results of datasets constructed in Jan 2025, Apr 2025, Jul 2025. This can observe the effectiveness of the data construction method in the time line."
>
> Our snapshots of generated benchmarking data **span more than half a year** in 2025 from January to July, which we believe could already provide valuable support for the robustness of our automatic benchmark construction and rigorous model evaluation/understanding therein along the time line. In the paper, we rigorously tested close to **30** open-source and closed-source LLMs on sample snapshots of our datasets to demonstrate the rigorous findings of OKBench, such as small LMs showing smaller gaps to large LMs on fresh knowledge when retrieval is in play. Having a denser evaluation with a timeline on a continuous stream of the benchmark is not our main focus, but can be an interesting avenue for future work.

---

> ### Author Response · Authors · 2025-12-03
>
> Below we answer the questions one by one:
>
> "1. As the questions are new (after the LLM is pre-trained), why some LLMs can still achieve as good performance as 50%?"
>
> This observation is very reasonable, and we are delighted to extend an explicit discussion of this limitation. We had an in-depth discussion regarding this matter while building the pipeline. First, models often have reasonable assumptions regarding the answer based on its parametric knowledge, especially in the multi-choice scenario. Taking the last multi-choice question of Table 3 as an example,
> > "As of March 22, 2025, which journal published the study findings on March 19 that detailed the impact of gantenerumab on delaying Alzheimer’s symptoms? A. The Lancet Psychiatry; B. JAMA Neurology; C. Neurology; D. The Lancet Neurology",
>
> based on the model's internal knowledge about these journals and differences between psychiatry/neurology, it might be able to pick out some unreasonable choices, resulting in a baseline accuracy higher than 25%. While it's unrealistic to completely remove these effects, we have explicitly provided instructions for our LM agents to generate choices that depend as large on ground truth articles as possible. Furthermore, such prior bias for guessing correct answers differs from model to model, which is hard to fully eliminate without heavy human involvement for every model for a fully automatic on-demand pipeline. That said, we totally agree that a more thorough contamination analysis and explicit discussion should be provided in the main text, and we will add it to the camera-ready version accordingly. We also recommend checking the supplementary results of the same set of models and retrievers on open-ended questions from Appendix H.
>
> "2. Is the following paper, which use new knowledge from Wikidata for evaluating LLMs, relevant? https://arxiv.org/abs/2412.17032"
>
> Although MINTQA is relevant in the broad landscape of evaluating LLMs on temporally new information, it targets a different problem setting from OKBench. As described in the paper, MINTQA constructs multi-hop reasoning chains over structured knowledge graph updates, and evaluates LLMs on their abilities to perform temporal integration and composition. We want to clarify that although OKBench is a dynamic benchmark, we don't focus on temporal abilities of LLMs as we don't assume the facts extracted from news necessarily evolve in time, which is the fundamental difference between our work and a series of temporal benchmarks based on Wikipedia. Our questions are extracted from full news passages released, ideally after the pretraining cutoff to test (small) language models' abilities to understand the provided context in a typical RAG scenario, whereas MINTQA uses post-2021 Wikidata edits as its knowledge source and generates reasoning paths from the KG graph.
>
> "3. How do you check the licenses or copyright of the news data providers? Does the news data include any sensitive personal information?"
>
> We believe we are not violating any policies of the news data providers in Table 2, but would be happy to discuss with the chairs, and revise our data collection practices if any issue is found that we may have missed.

---

### Official Review · Reviewer_wKmq · 2025-10-27

**Soundness:** 4
**Presentation:** 3
**Contribution:** 3
**Rating:** 6
**Confidence:** 5

**Summary:**

This paper introduces a novel framework for dynamic and decentralized evaluation of LLMs on evolving factual knowledge. The core contribution is OKBench, a fully automated, agentic pipeline that continuously generates, validates, and versions QA benchmarks from daily news streams. OKBench operates entirely autonomously, scraping fresh news articles, generating multiple-choice and open-ended questions with LLM agents, validating them through a second model, and assigning reproducible dataset signatures that ensure transparent version control.

The authors evaluate OKBench on both data quality and benchmark utility. Using OKBench, the paper conducts large-scale experiments across multiple open-source LLM families (e.g., Gemma, LLaMA, Qwen, Phi) and retrieval strategies (BM25, DPR, ColBERT v2).

**Strengths:**

The paper addresses an important and timely problem: the challenge of evaluating LLMs on factual knowledge without contamination from training data. The authors convincingly argue that testing on newly emerging knowledge is one of the most effective ways to mitigate data leakage, providing a well-motivated and practically relevant setting.

A key strength lies in the design of a fully autonomous benchmark generation pipeline, which makes OKBench highly scalable and easily reproducible. The inclusion of a data versioning protocol with unique dataset signatures is an especially thoughtful feature, ensuring transparent and repeatable evaluations across time.

The paper also demonstrates strong attention to data quality, supported by a human validation study confirming that the majority of automatically generated questions are clear and factually correct. Furthermore, the authors provide a detailed cost analysis, showing that daily benchmark generation is affordable, an important consideration for community adoption.

Finally, the experimental section is comprehensive and insightful, covering a broad range of models and retrieval-augmented generation methods, and yielding meaningful findings on knowledge freshness and retrieval effectiveness.

**Weaknesses:**

OKBench's novelty claim as “the first fully automated factual QA benchmark” is somewhat overstated. Previous works such as TemporalWiki[1], WikiFactDiff[2], and especially WikiBigEdit[3] have already introduced fully automated benchmark generation pipelines for evaluating factual and temporal knowledge in LLMs. Although OKBench distinguishes itself by focusing on the news domain rather than structured knowledge graph updates, these earlier benchmarks should be explicitly acknowledged and included in Table 2, and the corresponding claim of being the first fully automated benchmark (end of Section 2) should be softened accordingly.

Second, the empirical results in Section 5.1 raise concerns about data contamination, which directly undermines the benchmark’s stated motivation. If the benchmark genuinely captures unseen, post-cutoff knowledge, models should not substantially exceed the 25 % random baseline in the no-context setting. However, the reported results show significantly higher accuracies, suggesting that a nontrivial portion of the data overlaps with the models’ pretraining corpora or with frequently reported background facts. A more thorough contamination analysis or explicit discussion of this limitation would strengthen the paper’s credibility.

Third, the framework’s dependence on the underlying LLM used for question generation and validation is not explored experimentally. Since the agentic pipeline relies heavily on a single base model (GPT-4.1-2025-04-14), an ablation study varying the generation LLM would clarify the benchmark's robustness and reproducibility across different base models. Similarly, while the qualitative validation study is informative, a quantitative summary of the filtering rate (how many generated questions are discarded during validation) would help assess the pipeline’s efficiency and the actual yield of high-quality questions.

Finally, there are a few presentation issues that could improve readability: Figure 2 is introduced early but only discussed in Section 5.2, making its placement suboptimal. Figure 4 would benefit from adopting the same layout as Figure 3 (four subplots in a single row) with a less prominent color for the no-context baseline (this is more of a suggestion than an actual weakness).

[1] Jang et al. (2022): TemporalWiki: A lifelong benchmark for training and evaluating Ever-Evolving language models.
[2] Khodja et al. (2024): Wikifactdiff: A large, realistic, and temporally adaptable dataset for atomic factual knowledge update in causal language models.
[3] Thede et al. (2024): WikiBigEdit: Understanding the Limits of Lifelong Knowledge Editing in LLMs

**Questions:**

1) Novelty and Relation to Prior Work: How does OKBench fundamentally differ from WikiBigEdit, which also features a fully automated factual QA generation pipeline? Could the authors clarify whether they view OKBench as complementary to or extending this line of work, and why it was omitted from Table 2 and the related work section?
2) Data Contamination Analysis: Given that models achieve substantially higher than random accuracy in the no-context setting, how do the authors explain this performance?
3) Pipeline Robustness: Since the benchmark generation heavily relies on a specific base LLM (GPT-4.1-2025-04-14), how stable is the pipeline when using different generation or validation models?
4) Filtering Statistics: Can the authors quantify how many of the initially generated questions are filtered out during the validation stage?

---

> ### Author Response · Authors · 2025-12-03
>
> Dear reviewer wKmq,
>
> Thank you for your detailed and insightful review! We appreciate the chance to address each of your concerns below:
>
> "OKBench's novelty claim as “the first fully automated factual QA benchmark” is somewhat overstated. Previous works such as TemporalWiki[1], WikiFactDiff[2], and especially WikiBigEdit[3] have already introduced fully automated benchmark generation pipelines for evaluating factual and temporal knowledge in LLMs. Although OKBench distinguishes itself by focusing on the news domain rather than structured knowledge graph updates, these earlier benchmarks should be explicitly acknowledged and included in Table 2, and the corresponding claim of being the first fully automated benchmark (end of Section 2) should be softened accordingly.
>
> [1] Jang et al. (2022): TemporalWiki: A lifelong benchmark for training and evaluating Ever-Evolving language models.
>
> [2] Khodja et al. (2024): Wikifactdiff: A large, realistic, and temporally adaptable dataset for atomic factual knowledge update in causal language models.
>
> [3] Thede et al. (2024): WikiBigEdit: Understanding the Limits of Lifelong Knowledge Editing in LLMs"
>
> We are aware of this line of work focusing on temporal knowledge evaluation based on Wikipedia snapshots, which is orthogonal to our goal of evaluating retrieval augmented generation fairly based on fresh knowledge. Compared to the Wikipedia based temporal knowledge evaluation, our benchmark is **not specifically limited to the same set of knowledge and questions only with varying answers**, which was still a somewhat simple and “static” design. Instead, we focus on grounding **the emerging information in truly dynamic new events** happening at any moment. With that said, we agree that our claim of being “the first fully automated factual QA benchmark” may cause confusion. We will soften this statement at the end of Section 2 to be "the first fully on-demand automated factual QA benchmark" to capture our novelty more accurately, and explicitly add TemporalWiki, WikiFactDiff, and WikiBigEdit to Table 2 and the related work.
>
> "Second, the empirical results in Section 5.1 raise concerns about data contamination, which directly undermines the benchmark’s stated motivation. If the benchmark genuinely captures unseen, post-cutoff knowledge, models should not substantially exceed the 25% random baseline in the no-context setting. However, the reported results show significantly higher accuracies, suggesting that a nontrivial portion of the data overlaps with the models’ pretraining corpora or with frequently reported background facts. A more thorough contamination analysis or explicit discussion of this limitation would strengthen the paper’s credibility."
>
> This observation is very reasonable, and we are delighted to extend an explicit discussion of this limitation. We had an in-depth discussion regarding this matter while building the pipeline. First, models often have reasonable assumptions regarding the answer based on its parametric knowledge, especially in the multi-choice scenario. Taking the last multi-choice question of Table 3 as an example,
>
> > "As of March 22, 2025, which journal published the study findings on March 19 that detailed the impact of gantenerumab on delaying Alzheimer’s symptoms? A. The Lancet Psychiatry; B. JAMA Neurology; C. Neurology; D. The Lancet Neurology",
>
> based on the model's internal knowledge about these journals and differences between psychiatry/neurology, it might be able to pick out some unreasonable choices, resulting in a baseline accuracy higher than 25%. While it's unrealistic to completely remove these effects, we have explicitly provided instructions for our LM agents to generate choices that depend only on ground truth articles as much as possible. Furthermore, such prior bias for guessing correct answers differs from model to model, and it is not the stance of this paper to fully eliminate these effects for every model (and even if we can control it for every current model, the heavy engineering process will not be futureproof as new models will be built), which requires heavy human involvement. Nevertheless, in our design we make our best efforts to strike a balance between making the questions hard to guess and making the pipeline fully automatic. That said, we totally agree that a more thorough contamination analysis and explicit discussion should be provided in the main text, and we will add it to the camera-ready version accordingly. We also recommend checking the supplementary results of the same set of models and retrievers on open-ended questions from Appendix H, where the correct answers to new knowledge without the answer candidates are much harder to guess correctly.

---

> > ### Author Response · Authors · 2025-12-03
> >
> > "Third, the framework’s dependence on the underlying LLM used for question generation and validation is not explored experimentally. Since the agentic pipeline relies heavily on a single base model (GPT-4.1-2025-04-14), an ablation study varying the generation LLM would clarify the benchmark's robustness and reproducibility across different base models. Similarly, while the qualitative validation study is informative, a quantitative summary of the filtering rate (how many generated questions are discarded during validation) would help assess the pipeline’s efficiency and the actual yield of high-quality questions."
> >
> > We agree that understanding how OKBench behaves under different generation and validation LLMs is important for assessing its robustness. The pipeline should be model-agnostic conceptually, but similar to other previous agentic benchmarks, we relied on a single model version (GPT-4.1-2025-04-14) for consistency. We will add an ablation study in the appendix to compare different LLMs as generation agents. As for filtering rate, here are the statistics we gathered in the March-22nd-2025 snapshot of OKBench:
> >
> > |                              | Initial generation | Passed validation | Discarded |
> > |------------------------------|--------------------|-------------------|-----------|
> > | # of multiple-choice questions | 2350               | 2161              | 189       |
> > | Percentage                   | —                  | 91.96%            | 8.04%     |
> >
> > "Finally, there are a few presentation issues that could improve readability: Figure 2 is introduced early but only discussed in Section 5.2, making its placement suboptimal. Figure 4 would benefit from adopting the same layout as Figure 3 (four subplots in a single row) with a less prominent color for the no-context baseline (this is more of a suggestion than an actual weakness)."
> >
> > Thank you for these constructive suggestions to improve the visual presentation of our work! We are happy to modify these figure placements and colors accordingly and will make adjustments for the final version.

---

> ### Author Response · Authors · 2025-12-03
>
> Below are our answers to the questions list, one by one:
>
> "1. Novelty and Relation to Prior Work: How does OKBench fundamentally differ from WikiBigEdit, which also features a fully automated factual QA generation pipeline? Could the authors clarify whether they view OKBench as complementary to or extending this line of work, and why it was omitted from Table 2 and the related work section?"
>
> Thank you for raising this point. We will add WikiBigEdit to Table 2 and the related work section. At the same time, OKBench fundamentally targets a different problem setting than WikiBigEdit, and we view the two benchmarks as complementary rather than overlapping. WikiBigEdit is designed explicitly for lifelong knowledge editing: as outlined in Figure 1 of the WikiBigEdit paper, the pipeline compares consecutive Wikidata snapshots, identifies changed and unchanged triples, and generates rephrased question-answer pairs. This fits more into the line of work that investigates time-sensitive knowledge. Instead of time sensitivity of evolving knowledge, OKBench does not restrict to the overlapping set of knowledge, but focuses on more dynamic knowledge emerging from any moment, fairly revealing the benefits of retrieval augmentation over parametric memory, especially for small-sized language models.
>
> "2. Data Contamination Analysis: Given that models achieve substantially higher than random accuracy in the no-context setting, how do the authors explain this performance?"
>
> Please refer to the answer we provided under weaknesses in previous entries.
>
> "3. Pipeline Robustness: Since the benchmark generation heavily relies on a specific base LLM (GPT-4.1-2025-04-14), how stable is the pipeline when using different generation or validation models?"
>
> Please refer to the answer we provided under weaknesses in previous entries.
>
> "4. Filtering Statistics: Can the authors quantify how many of the initially generated questions are filtered out during the validation stage?"
>
> Please refer to the answer we provided under weaknesses in previous entries.

---

### Official Review · Reviewer_Ub36 · 2025-10-30

**Soundness:** 2
**Presentation:** 2
**Contribution:** 1
**Rating:** 2
**Confidence:** 4

**Summary:**

This work proposes OKBench as a fully automated framework for generating dynamic benchmarks.  It can be automatically generated and used for the evaluation of retrieval-augmented methods. On experiments with multiple open-source and proprietary LLMs, it finds multiple observation about the model behaviors toward novel information.

**Strengths:**

1. This work proposes an approach to construct an on-demand knowledge base from the Internet text source. It can be used for evaluation of RAG systems.

2. This work experiments LLMs’ behaviors on the dynamic corpus and discovers multiple intriguing observations.

**Weaknesses:**

1. The dataset novelty is limited.
- As summarized in Table 1, the novelty of the dataset is not clear compared to the existing ones.
- One key novel feature is “any time” in the update frequency, but making other datasets anytime too is not challenging.
- As such, no prominent novel features can be found for this dataset.

2. The novelty of the proposed data collection pipeline is limited.
- As illustrated in Fig. 1 and section 3.1, there is no novelty in the benchmark construction pipeline, as it is quite standard and straightforward.

3. The usability of the dataset is limited.
- As shown in Fig. 3, once the oracle document is given, most models attain very high accuracy (near 100%).
- It may implicate that once the retrieval is successful, the QA part could be very easy to solve. Then, this benchmark may evaluate mostly the retriever’s performance rather than LLMs’ QA capability.
- As reported in Fig.4, four different LLMs show almost similar performance. It could be a piece of evidence that the choice of an LLM does not matter to solve this benchmark. Only retriever selection matters.

4. Only three basic retrievers are tested.

**Questions:**

Please refer to the Weaknesses.

**Details Of Ethics Concerns:**

The data sources (news articles) summarized in Table 2 are under copyright. This work mentions fair-use but does not obtain permission from them. The proposed dataset construction pipeline may be under a risk of copyright infringement.

---

> ### Author Response · Authors · 2025-12-03
>
> Dear reviewer Ub36,
>
> We appreciate the opportunity to address each of your concerns below.
>
> "1. The dataset novelty is limited.
> - As summarized in Table 1, the novelty of the dataset is not clear compared to the existing ones.
> - One key novel feature is “any time” in the update frequency, but making other datasets anytime too is not challenging.
> - As such, no prominent novel features can be found for this dataset."
>
> Making a benchmark "any time" is not trivial, especially for these existing dynamic datasets we mentioned in Table 1. Benchmarks such as StreamingQA, RealTimeQA, and FreshQA rely on manual filtering, human-written questions, or fixed external sources, and their update cycles depend heavily on human involvement. These dataset sizes are therefore small, and automatically scaling up the size of anytime datasets on-demand is non-trivial, which is part of our contribution with the agentic design. In contrast, OKBench provides full automation across the entire pipeline with zero human intervention, which is what makes OKBench fully on-demand to scale up from the users' side. We agree this distinction deserves clearer emphasis and will revise Table 1 and Section 1 accordingly.
>
> "2. The novelty of the proposed data collection pipeline is limited.
> - As illustrated in Fig. 1 and section 3.1, there is no novelty in the benchmark construction pipeline, as it is quite standard and straightforward."
>
> We understand the concern that our figure and pipeline may look standard and straightforward.  However, the novelty is in the automation of every step of the pipeline.
>
> "3. The usability of the dataset is limited.
> - As shown in Fig. 3, once the oracle document is given, most models attain very high accuracy (near 100%).
> - It may implicate that once the retrieval is successful, the QA part could be very easy to solve. Then, this benchmark may evaluate mostly the retriever’s performance rather than LLMs’ QA capability.
> - As reported in Fig.4, four different LLMs show almost similar performance. It could be a piece of evidence that the choice of an LLM does not matter to solve this benchmark. Only retriever selection matters."
>
> You correctly observed that once the oracle article is provided, accuracy is high as long as the model is above some minimum size. This behavior is expected and intentional. OKBench is specifically designed to measure whether a model can incorporate newly acquired evidence, not to be difficult for models to answer (for example, requiring deep multi-hop reasoning on long documents). The moment the model is given the correct passage, the task reduces to short-span factual identification—similar to real-world RAG use cases. This is consistent with how most practical knowledge-intensive tasks are solved: if retrieval succeeds, QA becomes straightforward. However, we argue that the choice of LLMs still matters. As pointed out in Figure 4, not all of the LLMs get perfect accuracy when provided with context, especially in the case of smaller LLMs. To make it clearer, we recommend reading Appendix G with detailed result statistics, and we will improve the visualization of Figure 4 to make the differences clearer visually.
>
> "4. Only three basic retrievers are tested."
>
> Our intention was not to benchmark the strongest retrievers available, but to diagnose how different retrieval properties interact with OKBench’s design. For this purpose, the DPR vs. BM25 vs. ColBERTv2 contrast is already informative. We hope that OKBench can serve as a valuable resource to evaluate (small) models' abilities to utilize provided contexts, not to evaluate retrievers.
>
> "The data sources (news articles) summarized in Table 2 are under copyright. This work mentions fair-use but does not obtain permission from them. The proposed dataset construction pipeline may be under a risk of copyright infringement."
>
> We believe we are not violating any policies of the news data providers in Table 2, but would be happy to discuss with the chairs, and revise our data collection practices if any issue is found that we may have missed.

---

### Official Review · Reviewer_osHV · 2025-11-01

**Soundness:** 1
**Presentation:** 3
**Contribution:** 1
**Rating:** 2
**Confidence:** 4

**Summary:**

This paper introduces OKBench, a dynamic and knowledge-intensive benchmark designed to automatically evaluate large language models on their ability to handle factual, up-to-date information.

**Strengths:**

- This paper proposes OKBench, a dynamic, knowledge-intensive benchmark that is automatically updated.
- The authors test models of various sizes and compare their performance on MMLU Pro and OKBench, analyzing differences in memorization and adaptability to newly introduced information.

**Weaknesses:**

**Limited novelty:**
- Although the paper claims that OKBench is “the first fully automated benchmark for evaluating factual question answering ability” (L146), similar dynamic benchmarks have already been proposed in prior work [1, 2, 3].
- These existing works also describe pipelines for continual updates, whereas this paper presents only the initial construction pipeline, without demonstrating an actual update process. The lack of comparison to prior dynamic benchmarks may mislead readers about the paper’s contribution.
- The paper asserts that it introduces an agentic framework for benchmark construction (L153). However, the pipeline illustrated in Figure 2 lacks a clear agentic component; each step appears to be manually designed, without autonomous decision-making or agent-based iteration.


**Benchmark difficulty and saturation:**
- The results in Figure 4 show that BM25 Context performance (≈90–95 for Gemma) is nearly identical to Oracle performance (≈95). Even accounting for BM25 being a simple lexical retriever, this narrow gap suggests that the benchmark questions may be too easy.
- Moreover, BM25 Context significantly outperforms DPR Context (90–95 vs. 75–80), indicating that word matching alone suffices to answer most questions.
- This also implies that questions might have been generated from single factual sentences, often reusing phrases directly from the source text.
 Such design choices reduce the benchmark’s ability to evaluate deeper reasoning and may compromise its validity.

[1] Ko et al., "GrowOVER: How Can LLMs Adapt to Growing Real-World Knowledge?", ACL 2024.
[2] Lin et al., "DynaQuest: A Dynamic Question Answering Dataset Reflecting Real-World Knowledge Updates", ACL 2025 Findings.
[3] Ouyang et al., "HoH: A Dynamic Benchmark for Evaluating the Impact of Outdated Information on Retrieval-Augmented Generation", ArXiv 2025.

**Questions:**

- The DPR retriever shows much lower performance than BM25 or ColBERTv2, and inconsistent results across the 1-, 5-, and 10-day corpus.
Have the authors considered evaluating stronger retrievers, such as Qwen-3-Embedding or E5, which are known to perform better on factual retrieval tasks? Including such models could clarify whether the issue lies with the retriever’s capability or with the benchmark’s inherent design.

---

> ### Author Response · Authors · 2025-12-03
>
> Dear reviewer osHV,
> Thank you for the detailed and thoughtful review. We appreciate the chance to clarify our contributions and address your concerns point by point.
>
> **Regarding limited novelty:**
>
> "Although the paper claims that OKBench is “the first fully automated benchmark for evaluating factual question answering ability” (L146), similar dynamic benchmarks have already been proposed in prior work [1, 2, 3].
> These existing works also describe pipelines for continual updates, whereas this paper presents only the initial construction pipeline, without demonstrating an actual update process. The lack of comparison to prior dynamic benchmarks may mislead readers about the paper’s contribution.
> The paper asserts that it introduces an agentic framework for benchmark construction (L153). However, the pipeline illustrated in Figure 2 lacks a clear agentic component; each step appears to be manually designed, without autonomous decision-making or agent-based iteration.
>
> [1] Ko et al., "GrowOVER: How Can LLMs Adapt to Growing Real-World Knowledge?", ACL 2024.
>
> [2] Lin et al., "DynaQuest: A Dynamic Question Answering Dataset Reflecting Real-World Knowledge Updates", ACL 2025 Findings.
>
> [3] Ouyang et al., "HoH: A Dynamic Benchmark for Evaluating the Impact of Outdated Information on Retrieval-Augmented Generation", ArXiv 2025."
>
> Thank you for pointing out GrowOVER, DynaQuest, and HoH as important prior works. You are absolutely right that there are some works that explore timely evolving QA settings. However, we want to point out that our contribution differs from these efforts in a few key ways:
> 1. Objective: Prior work mainly studies how Wikipedia knowledge becomes outdated over time. The focus of our work is not on the timely evolving manner of knowledge, but on providing a fully automated factual benchmark that updates continuously from real-time news articles without human labeling.
> 2. Time-sensitive knowledge and update frequency: GrowOVER, DynaQuest and HoH all focus on Wikipedia knowledge updates instead of fresh knowledge. They do not incorporate freshly emerging information outside Wikipedia. GrowOVER takes Wikipedia snapshots once per month, and focuses on evaluating evolving knowledge across the snapshots. DynaQuest constructs temporal QA from infobox edits in Wikipedia and updates bi-weekly. HoH is also Wikipedia-based, updates monthly and focuses specifically on outdated-information harm in RAG (e.g., co-existing old vs. new evidence), not on real-time factual evaluation. In contrast, the data domain of OKBench is real-world news, and targets real-time facts that appear on a given day of publication, not necessarily evolved from past facts. This makes OKBench substantially different from these previous efforts in the type and freshness of knowledge it captures. Plus, OKBench includes an automated answer-verification stage and multi-source news extraction that can run from any user's side with no human post-processing. Plus, our update cycle is "anytime", which means users could get a version-labeled snapshot of the benchmark instantly by running our pipeline, unlike prior works that rely partially on human efforts to update.
> 3. Agentic framework: we respectfully disagree with the reviewer comment. To fulfill the vision that a truly dynamic knowledge benchmarking system can be created **by anyone at any time on-demand**, OKBench is designed exactly with the goal of **full automation**, where every component in the construction pipeline is built from a series of LLM-driven workflows. Figure 2 illustrated the **fully automated pipeline**, which can **run by itself** and does not involve human intervention. Benchmark quality is controlled by the data generating agent and the verification agent, which can run iteratively depending on the budget, and data distribution is managed by an automatic versioning system. Therefore, we do not understand the reviewer’s question regarding our agentic framework not being “agentic”, as our whole pipeline is clearly built with “autonomous decision-making” and “agent-based iteration”. Please see Reviewer wKmq’s review summary that also attests to the autonomous nature of our framework.
>
> (We believe the reviewer may have some misunderstanding of the term “agentic framework”. An agentic system is composed of autonomous modules that can make decisions/take actions, put together for a complex task. LLMs are a powerful tool to serve as these autonomous modules. And of course agentic systems are still designed by humans—at least now for most of the cases, which is the purpose of rigorously describing this and much other research.)

---

> > ### Author Response · Authors · 2025-12-03
> >
> > **Regarding benchmark difficulty and saturation:**
> >
> > "The results in Figure 4 show that BM25 Context performance (≈90–95 for Gemma) is nearly identical to Oracle performance (≈95). Even accounting for BM25 being a simple lexical retriever, this narrow gap suggests that the benchmark questions may be too easy.
> > Moreover, BM25 Context significantly outperforms DPR Context (90–95 vs. 75–80), indicating that word matching alone suffices to answer most questions.
> > This also implies that questions might have been generated from single factual sentences, often reusing phrases directly from the source text. Such design choices reduce the benchmark’s ability to evaluate deeper reasoning and may compromise its validity."
> >
> > We want to clarify that our goal is not to make the questions difficult, but to develop a benchmark that can fairly reveal the benefits of retrieval augmentation over parametric memory. For this purpose we want the minimal difficulty required to cause parametric memory to fail.
> >
> > **Regarding retriever choices:**
> >
> > "The DPR retriever shows much lower performance than BM25 or ColBERTv2, and inconsistent results across the 1-, 5-, and 10-day corpus. Have the authors considered evaluating stronger retrievers, such as Qwen-3-Embedding or E5, which are known to perform better on factual retrieval tasks? Including such models could clarify whether the issue lies with the retriever’s capability or with the benchmark’s inherent design."
> >
> > Thank you for raising this concern. We acknowledge that DPR and ColBERTv2 both show lower performance than BM25, which makes intuitive sense as our question type is factual QA and BM25 is a lexical retriever that works better in terms of people or event names.
> > Our goal in including retrieval baselines is not to benchmark the state of the art in retrieval, but to diagnose how different retrieval properties interact with OKBench’s design. For this purpose, the DPR vs. BM25 vs. ColBERTv2 contrast is already informative. OKBench aims to evaluate models, not retrievers. It would therefore be counterproductive to bake in assumptions tied to the very latest embedding models which may change rapidly or require proprietary access. RAG models may vary widely in how well they use retrieved context, thus introducing a much stronger dense model does not change the qualitative pattern we analyze.

---

### Author Response · Authors · 2025-12-03
**General Response**

To the AC,

We would like to summarize the key issues raised in the reviews and our responses.  First we respectfully note that the reviews appear to be rather low-quality as a whole, in the sense that they include comments that are unsupported with any details and seem incorrect or at least perplexing (example: “The current evaluation … ignore the dataset construction method, and the key features of the dataset”, without stating what additional details are missing).  It is unfortunate that we do not have the opportunity to receive an additional response from the reviewers (which we realize is in part due to our own late submission of the responses).

The main issues raised by reviewers are:

- Insufficient novelty relative to existing dynamic factual QA benchmarks.  We agree that we should add several new benchmarks to our related work section, and are grateful to the reviewers for pointing them out.  However, our work differs significantly from these existing benchmarks.  These prior benchmarks are characterised by either or both (1) required human involvement or (2) dependence on structured knowledge extracted from Wikipedia.  As far as we know, our benchmark is the first factual QA benchmark that is (1) fully automated and (2) based on extracting QA pairs from natural language news stories.  None of these prior approaches can generate a quiz on demand about, say, today’s news.  It is not at all trivial to design a fully automatic pipeline for this purpose.

- Performance of existing LLMs is too good.  We respectfully disagree.  We designed the benchmark to be (1) easy for a retrieval-augmented model but (2) difficult without retrieval.  We have achieved this:  There is a large gap in performance between the worst-case (no retrieval) and best-case (oracle retrieved context) settings.  It is true that the worst-case performance is significantly better than random (i.e., >25% accuracy for our 4-way multiple-choice questions).  This is largely because the set of distractor choices vary in how easy or difficult it is to exclude them, and to a lesser extent because it is impossible to completely remove all cases of a news story reporting facts that predate the story.  It is very difficult for a multiple-choice test of this kind to have random baseline performance.  It is also not necessary to have random baseline performance:  As long as there is a sizeable gap between no-retrieval and with-retrieval, the benchmark is satisfying its goals.  We agree that we should, and we will, clarify in the paper these potential sources of above-random baseline performance.

There are other concerns and questions about lower-level details raised by the reviewers, which we leave to the individual responses.

---

### Meta-Review · Area_Chair_XroW · 2026-01-06

**Summary:**

This paper presents an end-to-end automated pipeline for benchmark construction, resulting in OpenKnowledgeBench (OKBench), a dynamic knowledge benchmark. However, reviewers raised concerns regarding the strength of the novelty claim, noting the existence of prior dynamic benchmarks, automated benchmarks, and works that combine both aspects. In addition, the benchmark itself was viewed as a limited contribution: the questions may be overly easy and appear to evaluate retriever effectiveness rather than deeper reasoning capabilities.

**Reviewer Concerns:**

In the rebuttal, the authors acknowledged missing related work and clarified that OKBench specifically targets real-time news. They also argued that the high accuracy observed under retrieval settings is an intentional design choice. Nevertheless, to address the reviewers’ concerns, the authors would need to invest substantial effort in precisely articulating their core contributions and clearly positioning OKBench relative to existing benchmarks. This is a nontrivial revision rather than a marginal clarification.

**Reviewer Scores:**

I expect that the reviewers will maintain their original scores.

---

### Decision · Program_Chairs · 2026-01-26

Reject